# Many Experiments, Few Repetitions, Unpaired Data, and Sparse Effects: Is Causal Inference Possible?

**Felix Schur** [1]  **Niklas Pfister** [2]  **Peng Ding** [3]  **Sach Mukherjee** [4][5]  **Jonas Peters** [1]

## Abstract

In many applications, practical constraints prevent measuring covariates and outcomes on the same units, resulting in unpaired data. We study the problem of estimating causal effects under hidden confounding in the following unpaired data setting: we observe some covariates $X$ and an outcome $Y$ under different experimental conditions (environments) but do not observe them jointly – we either observe $X$ or $Y$. Under appropriate regularity conditions, the problem can be cast as an instrumental variable (IV) regression with the environment acting as a (possibly high-dimensional) instrument. When there are many environments but only a few observations per environment, standard two-sample IV estimators fail to be consistent. We propose a GMM-type estimator (SPLITUP) based on cross-fold sample splitting of the instrument–covariate sample that also applies in standard IV settings. We prove that it is consistent as the number of environments grows but the sample size per environment remains constant. We further extend the method to sparse causal effects via $\ell_1$-regularized estimation and post-selection refitting.

## 1. Introduction

Suppose we have a response $Y$ (e.g., a phenotype), some covariates $X$ (e.g., a genotype) and that we observe the system under $m$ different experimental conditions (e.g., using gene modification technology). Suppose we are interested in estimating the causal effect from $X$ to $Y$ but that

[1]Department of Mathematics, ETH Zurich, Zurich, Switzerland [2]Lakera AI, Zurich, Switzerland [3]Department of Statistics, University of California, Berkeley, CA, USA [4]German Center for Neurodegenerative Diseases (DZNE) & University of Bonn, Bonn, Germany [5]MRC Biostatistics Unit, University of Cambridge, Cambridge, United Kingdom. Correspondence to: Felix Schur <felix.schur@stat.math.ethz.ch>.

*Proceedings of the 43$^{rd}$ International Conference on Machine Learning*, Seoul, South Korea. PMLR 306, 2026. Copyright 2026 by the author(s).

we are facing the following two challenges: In each experiment, we may observe multiple i.i.d. repetitions but for each repetition, we can either measure $X$ or $Y$ but not both (see Table 1) and there may be unobserved confounding

| Exp. | 1 | 1 | $\cdots$ | 2 | 2 | 2 | $\cdots$ | $m$ |
|------|------|------|----------|------|------|------|----------|------|
| $X_1$ | 1.8 | $\times$ | $\cdots$ | 3.2 | $\times$ | $\times$ | $\cdots$ | 0.6 |
| $\vdots$ | $\vdots$ | $\vdots$ | $\vdots$ | $\vdots$ | $\vdots$ | $\vdots$ | $\vdots$ | $\vdots$ |
| $X_d$ | 2.0 | $\times$ | $\cdots$ | 0.2 | $\times$ | $\times$ | $\cdots$ | 1.0 |
| $Y$ | $\times$ | 2.5 | $\cdots$ | $\times$ | $-3.7$ | 1.1 | $\cdots$ | $\times$ |

*Table 1.* We consider the problem of estimating the causal effect between $X$ and $Y$ in the presence of hidden confounding when data are unpaired. The table shows an example dataset, where we observe the system under $m$ different experimental conditions.

between $X$ and $Y$. It may come as a surprise that in such a setting, consistent estimation of causal effects is possible under weak assumptions.

**IV and regression of the means.**  These assumptions are satisfied, e.g., if the experimental conditions influence $X$ but do not influence $Y$ directly or via another causal path (we will make this precise in Section 3). This condition is well-studied in the literature on instrumental variables (IV) (Angrist and Krueger, 1992; Imbens and Angrist, 1994; Angrist et al., 1996; Staiger and Stock, 1997). Let us assume for a moment that for each experimental condition $j \in \{1, \ldots, m\}$, we observe all $X$ and all $Y$, so the data are paired. We can then apply IV estimators to the example above when using a one-hot encoding for the different experimental conditions. In this case (see Appendix D for a derivation), the classical two-stage least squares (TSLS) estimator corresponds to the following simple procedure: within each experimental condition, we separately average the $X$'s and $Y$'s, resulting in a single (average) value for $X$ and a single value for $Y$ per experiment; we then regress the $m$ values containing $Y$ averages on the $m$ values containing $X$ averages. Clearly, this procedure does not rely on the $X$ and $Y$ values to be paired, so it is still applicable in the unpaired setting. This then is not an instantiation of IV anymore but of two-sample IV (TS-IV) (Angrist and Krueger, 1992; Inoue and Solon, 2010; Burgess et al., 2013), a modification of IV, which has been developed for unpaired data (without a special focus on categorical instru-

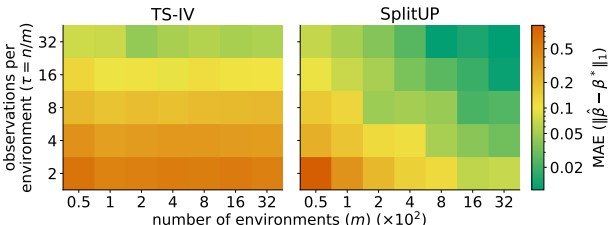

*Figure 1.* Estimating causal effects in a data set of the form of Table 1. Our proposed estimator SPLITUP (right) outperforms existing methods (two-sample IV (TS-IV) on the left) if the number $m$ of experimental conditions increases. Indeed, we prove that unlike existing estimators (such as TS-IV on the left), SPLITUP is consistent as $n, m \to \infty$ and $n/m \to r \in (0, \infty)$. The data for this figure is generated as described in Section 5 (Setting 2) with 10 independent runs.

ments). We regard these two algebraically trivial insights as important as the simple estimator of taking the average per experiment and then performing regression on the average is often considered in practice – both in paired and unpaired settings (Deaton, 1985; Greenland and Longnecker, 1992; King et al., 2004; Peres-Neto et al., 2017). The reasoning above provides a precise interpretation of the target of inference: if the experimental condition does not modify the causal effect, the averaging allows us to remove hidden confounding.

**More repetitions versus more observations.** Applying standard theory from instrumental variables now provides us with guarantees such as consistent estimation if the number of observations per experiment converges to infinity. In practice, however, we may instead see many experiments and few repetitions per experiment, a setting that is not well-covered by these asymptotics. In this work, we propose a different estimator (SPLITUP) that targets a setting, where the number $m$ of experimental conditions (i.e., the number of instruments) is growing but the number of repetitions per experiment remains constant. We prove identifiability in the setting with $m \to \infty$ and prove consistency of the estimator. Figure 1 shows that SPLITUP outperforms classical estimators such as TS-IV on finite data.

**Motivation from biology.** Applications in which data are obtained in an unpaired fashion under different causal environments abound, particularly in the biomedical domain. In contemporary biology, it is possible to measure molecular variables (such as gene or protein expression levels) at large scale and under different causal regimes (such as defined interventions at the gene or protein level or via chemical compounds) and such data are the focus of much ongoing work in biotechnology and machine learning (see, e.g., Replogle et al., 2022; Lopez et al., 2022; Lagemann et al., 2023). Furthermore, it is increasingly common to consider many experimental conditions (e.g. intervention on many

different genes). Applied interest often focuses on the effects of such interventions on concrete phenotypes (such as growth rate or cellular behaviour) and it is therefore common to study phenotypes under intervention ('screening'). However, cells may have to be disturbed or destroyed in order to obtain molecular measurements, hence in studies in which both gene/protein levels $X$ and phenotypes $Y$ are of interest, the actual cells involved may be different and the data are therefore unpaired in the sense of this paper. These issues are also important in molecular medicine, where interventions on laboratory cells are used to shed light on patient responses to the same interventions (such as studying the effect of multiple cancer drugs on cells in the lab and linking these data to treatment outcomes (Yang et al., 2012; Kirkham et al., 2025)). In such settings, the data are typically unpaired since the actual cells involved in the various steps are distinct.

**Continuous instruments and many covariates.** The estimators and theoretical results that we develop in this paper apply to continuous instruments, too. We also allow for settings with many covariates: identifiability and consistency still hold when the number of covariates is strictly larger than the number of instruments – if the causal effect can be assumed to be sparse (see Section 3 and Section 4).

## 1.1. Related Work

**Two-sample IV and Mendelian randomization.** Our unpaired IV setting is connected to two-sample IV estimators that combine instrument–exposure and instrument–outcome moments across samples (Angrist and Krueger, 1995; Inoue and Solon, 2010; Pacini and Windmeijer, 2016; Zhao et al., 2019; Angrist and Krueger, 1992) and to two-sample (summary-data) Mendelian randomization (Hartwig et al., 2016; Zhao et al., 2020). This literature also highlights practical biases from winner's curse and sample overlap (Pierce and Burgess, 2013; Burgess et al., 2016) and robustness to horizontal pleiotropy (Hartwig et al., 2017; Qi and Chatterjee, 2019; Morrison et al., 2020). In contrast, we have settings in mind with large $m$ and $d$; theoretically, we focus on consistent estimation in regimes with high-dimensional instruments ($n/m \to c \in (0, \infty)$) and sparse causal effects, where naive two-sample combinations can fail (see Section 4.2).

**High-dimensional, sparse, and many-instrument IV.** High-dimensional IV under sparsity has been studied via two-stage regularization (Zhu, 2018; Lin et al., 2015; Chen et al., 2018), debiasing/orthogonal moments and post-selection inference (Gold et al., 2020; Belloni et al., 2022), identifiability-driven estimators in paired data (Pfister and Peters, 2022; Tang et al., 2023), and summary-data adaptations (Huang et al., 2024). Related approaches regularize

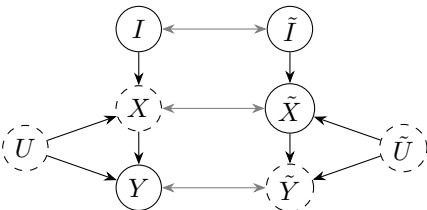

*Figure 2.* Simplified visualization of the data generating process in Eq. (2.1). $U$ and $\tilde{U}$ represent unobserved confounders that visualize the dependence between $X$ and $\varepsilon$. The model allows for complex dependencies (and even equalities) between variables in the tilde and non-tilde world, as indicated by the gray edges. We discuss other, including more general data-generating processes that satisfy (2.1) in Appendix B.

GMM objectives directly (Caner, 2009; Shi, 2016; Fan and Liao, 2014). Our analysis also relates to weak- and many-instrument asymptotics and diagnostics (Staiger and Stock, 1997; Bekker, 1994; Donald and Newey, 2001; Hansen et al., 2008; Stock and Yogo, 2005; Andrews et al., 2019; Belloni et al., 2012; 2014) and to weak-instrument robust testing in two-sample IV (Choi et al., 2018). We show that, for unpaired observations in the many-instrument regime, naive estimators incur a measurement-error-type bias absent from paired-sample IV, and we provide bias-corrected consistent (and, under suitable conditions, asymptotically normal) estimators.

An extended version of the related work (with additional comparisons and discussion) is in Appendix A.

## 2. Unpaired Data with Hidden Confounding

We consider two triplets of random variables: $(I, X, Y)$ and $(\tilde{I}, \tilde{X}, \tilde{Y})$ but assume access only to partial observations. Specifically, we observe $(I, Y)$ and $(\tilde{I}, \tilde{X})$ (or, more precisely, multiple i.i.d. copies thereof), the variables $X$ and $\tilde{Y}$ remain unobserved. The corresponding data-generating process is given by (unobserved variables are shown in gray):

$$Y = X^\top \beta^* + \varepsilon, \qquad \tilde{Y} = \tilde{X}^\top \tilde{\beta}^* + \tilde{\varepsilon}, \qquad (2.1)$$

where $\beta^*$ and $\tilde{\beta}^*$ represent the causal effects, that is, $\mathbb{E}[Y \mid \mathrm{do}(X := x)] = x^\top \beta^*$, $\mathbb{E}[\tilde{Y} \mid \mathrm{do}(\tilde{X} := x)] = x^\top \tilde{\beta}^*$.

The noise terms $\varepsilon$ and $\tilde{\varepsilon}$ are not assumed to be independent of $X$ and $\tilde{X}$, respectively (thereby, implicitly, allowing for hidden confounding between $X$ and $Y$. Our linearity assumption concerns only the structural effect of $X$ on $Y$; the dependence of $X$ on the instrument and the dependence induced by hidden confounding may be nonlinear. Figure 2 shows a graphical representation of the data-generating process. The setting even allows for dependence between the two samples. We discuss other data-generating processes that satisfy (2.1) in Appendix B.

In Section 3, we show that the causal effect $\beta^*$ (or $\tilde{\beta}^*$ or both) is identifiable. This is non-trivial because of the following reasons: (i) Due to the hidden confounding, naively regressing $Y$ on $X$ (or $\tilde{Y}$ on $\tilde{X}$) (if these were available) would generally lead to biased estimates. (ii) We only have unpaired data, so regressing $Y$ on $X$ (or $\tilde{Y}$ on $\tilde{X}$) is not feasible. Unlike classical work on unpaired data or two-sample IV $(I, X, Y)$ and $(\tilde{I}, \tilde{X}, \tilde{Y})$ may but do not have to be equal in distribution. (iii) Observations across the two samples may be dependent, see, e.g., Appendix B.

### 2.1. Setting and Main Assumptions

Let $n, \tilde{n}, m, d \in \mathbb{N}$. Let $(I, X, Y)$ and $(\tilde{I}, \tilde{X}, \tilde{Y})$ be random vectors, whose components are $m$, $d$, and 1-dimensional, respectively, and which generated by the process described in (2.1) where $\beta^*, \tilde{\beta}^* \in \mathbb{R}^d$ and $\varepsilon$ and $\tilde{\varepsilon}$ are random variables. We assume that all random variables have finite second moments. We assume that there exist two i.i.d. samples $\{(I_i, X_i, Y_i)\}_{i \in [n]}$ and $\{(\tilde{I}_i, \tilde{X}_i, \tilde{Y}_i)\}_{i \in [\tilde{n}]}$, but the observed data are *unpaired*: in the first system we only observe $\{(I_i, Y_i)\}_{i \in [n]}$ and in the second $\{(\tilde{I}_i, \tilde{X}_i)\}_{i \in [\tilde{n}]}$. The case $(I, X, Y) \overset{d}{=} (\tilde{I}, \tilde{X}, \tilde{Y})$ is contained as a special case.

**Assumption 1.** *We have*

*(i)* $\mathrm{Cov}(\tilde{I}, \tilde{X}) = \mathrm{Cov}(I, X)$ *and*
*(ii)* $\mathbb{E}[\varepsilon \mid I] = 0$.

**Assumption 1'.** *We have*

*(i')* $\mathrm{Cov}(\tilde{I}, \tilde{Y}) = \mathrm{Cov}(I, Y)$ *and*
*(ii')* $\mathbb{E}[\tilde{\varepsilon} \mid \tilde{I}] = 0$.

We will see in Section 3 that under Assumption 1 we can identify $\beta^*$ in the non-tilde system and under Assumption 1' we can identify $\tilde{\beta}^*$ in the tilde system. In Appendix C we further show that Assumption 1 (Assumption 1', respectively) are not only sufficient, but also necessary for identifiability (in the non-degenerate, scalar case). We also provide a sensitivity analysis for violations of Assumption 1. To simplify notation, in this work, we focus on identifying $\beta^*$ using Assumption 1; analogous arguments hold when we identify $\tilde{\beta}^*$ using Assumption 1'. Assumption 1 (ii) (or Assumption 1' (ii')) is called exclusion restriction or exogeneity and is commonly exploited in the IV literature (Imbens and Angrist, 1994; Angrist et al., 1996; Angrist and Pischke, 2009); it allows us to identify the causal effect despite hidden confounding. Assumption 1 (i) (or Assumption 1' (i')) connects the two systems and guarantees that we can transfer information from one system to the other.

### 2.2. Environments as Instruments

In Section 1, we have argued that if we have access to data collected from multiple environments—for instance, from

different experimental setups or hospitals—but we lack explicit features describing these environments we can still treat the environment indicator as a vector of categorical instruments (via one-hot encoding). In such a setting, a sufficient condition for Assumption 1 (i) is that we observe $\tilde{X}$ and $Y$ under the same environment distribution, and that the conditional expectations of $X$ and $\tilde{X}$ coincide within each environment: $\mathbb{E}[X|I] = \mathbb{E}[\tilde{X}|I]$. An analogous condition holds for Assumption 1 (i').

## 3. Identifiability

We now show that, under Assumption 1, the causal effect $\beta^*$ is identifiable from the joint distributions of $(\tilde{I}, \tilde{X})$ and $(I, Y)$. Without loss of generality and to simplify notation, we assume that all variables are centered and that all variables have finite second moments. All proofs can be found in Appendix L.

We first consider the partially known *finite-dimensional instrument* setting (defined as the setting with $m$ fixed) in Section 3.1 and later, in Section 3.2, we consider the *high-dimensional instrument* setting (defined as the asymptotic setting with $m \to \infty$). In both settings, we consider $d$ to be fixed and differentiate the cases where $\beta^*$ is dense and where $\beta^*$ is sparse. Some remarks are in order. Although we rely on standard results for the Lasso (Tibshirani, 1996; Bühlmann and Van De Geer, 2011), the sparse $\beta^*$ setting is not 'high-dimensional' in the sense of $d \to \infty$ (even if the distribution of $(I, X, Y)$ is known, $\beta^*$ is, in general, not identified if $d > \mathrm{rank}(\mathrm{Cov}(I, X))$). In contrast, in the high-dimensional instrument setting, $d$ is fixed but $m \to \infty$ (and $n/m \to r \in (0, \infty)$). In this case, we assume that the limit $Q := \lim_{m \to \infty} m \, \mathrm{Cov}(\tilde{I}, \tilde{X})^\top \mathrm{Cov}(\tilde{I}, \tilde{X}) \in \mathbb{R}^{d \times d}$ exists and is well-defined.

### 3.1. Finite-Dimensional Instrument

Under Assumption 1 we have

$$\mathrm{Cov}(I, Y) - \mathrm{Cov}(\tilde{I}, \tilde{X})\beta^* = \mathrm{Cov}(I, Y) - \mathrm{Cov}(I, X)\beta^*$$
$$= \mathrm{Cov}(I, Y - X^\top \beta^*) = \mathbb{E}\left[I \mathbb{E}\left[\varepsilon \mid I\right]\right] = 0. \tag{3.1}$$

Here, we used Assumption 1 (i) in the first equation and Assumption 1(ii) in the last equation. Analogously, we can use Assumption 1' to get $\mathrm{Cov}(I, Y) - \mathrm{Cov}(\tilde{I}, \tilde{X})\tilde{\beta}^* = 0$. We now define the set of solutions $\mathcal{S}$ as

$$\mathcal{S} := \{\beta \in \mathbb{R}^d \mid \mathrm{Cov}(I, Y) = \mathrm{Cov}(\tilde{I}, \tilde{X})\beta\}$$

($\mathcal{S}$ can be computed from the joint distribution of $(\tilde{X}, \tilde{I})$ and the joint distribution of $(I, Y)$). We always have $\beta^* \in \mathcal{S}$. The causal effect $\beta^* \in \mathbb{R}^d$ is identifiable from the joint distribution of $(\tilde{X}, \tilde{I})$ and the joint distribution of $(I, Y)$ via these moment-conditions if and only if $\mathcal{S} = \{\beta^*\}$. We

now state a well-known result from the two-sample IV (also called two-sample MR) literature (Angrist and Krueger, 1995; Inoue and Solon, 2010; Hartwig et al., 2016).

**Proposition 3.1** (Identifiability for dense $\beta^*$). *Assume Assumption 1. We have* $\mathcal{S} = \{\beta^*\}$ *if and only if*

$$\mathrm{rank}(\mathrm{Cov}(I, X)) = d. \tag{3.2}$$

*In particular, $m \geqslant d$ is necessary for identifiability.*

Proposition 3.1 shows that we have identifiability in our general unpaired setting under the same conditions as for the paired IV setting. Next, we consider the sparse setting, i.e., $\|\beta^*\|_0 \leqslant s^* \leqslant d$. In this case, we may have identifiability even if $\mathrm{rank}(\mathrm{Cov}(I, X)) < d$. We now consider

$$\mathcal{S}_0 := \underset{\beta \in \mathcal{S}}{\arg\min} \|\beta\|_0.$$

Defining $\Sigma_t := \{v \in \mathbb{R}^d : \|v\|_0 \leqslant t\}$, we obtain the following identifiability result.

**Theorem 3.2** (Identifiability for sparse $\beta^*$). *Assume Assumption 1. The following statements are equivalent:*

*(i) (Identifiability via sparsest solution) For all $\beta^* \in \mathbb{R}^d$ with $\|\beta^*\|_0 \leqslant s^*$, it holds that $\mathcal{S}_0 = \{\beta^*\}$.*
*(ii) (Restricted nullspace) $\ker(\mathrm{Cov}(I, X)) \cap \Sigma_{2s^*} = \{0\}$.*

*In particular, $m \geqslant \min(d, 2s^*)$ is necessary for identifiability.*

Theorem 3.2 shows that the restricted nullspace condition for $\mathrm{Cov}(I, X)$ is a necessary and sufficient condition for identifiability via $\mathcal{S}_0$. The conditions for sparse identifiability are weaker than identifiability in the dense setting and they are strictly weaker if and only if $d \geqslant 2s^*$. Intuitively, because we want uniqueness among all $s^*$-sparse solutions, we need to ensure that the difference of two $s^*$-sparse vectors lies in the nullspace, and that difference can have up to $2s^*$ nonzeros, hence the 2. This shows that in settings where we can reasonably assume sparse causal effects in unpaired data, identifiability is still possible even with weak instruments (e.g., when $m$ is small). Pfister and Peters (2022) also study the identifiability question for sparse effects but in the context of causal graphs associated with the linear SCMs (Pearl, 2009; Bongers et al., 2021) and using penalization based on $\ell_0$ rather than $\ell_1$.

### 3.2. High-dimensional Instruments

In the high-dimensional instrument setting we assume that $m \to \infty$, so the dimensions of $\mathrm{Cov}(I, Y)$ and $\mathrm{Cov}(\tilde{I}, \tilde{X})$ increase with $n$. We consider the limit $Q = \lim_{m \to \infty} m \, \mathrm{Cov}(\tilde{I}, \tilde{X})^\top \mathrm{Cov}(\tilde{I}, \tilde{X})$ assuming that it exists; conditions for identifiability are then expressed in terms of $Q$ rather than the finite-

$m$ matrix[1]. Under Assumption 1, we have $Q_Y :=$ $\lim_{m \to \infty} m \operatorname{Cov}(\tilde{I}, \tilde{X})^\top \operatorname{Cov}(I, Y) = Q\beta^*$. Thus,

$$\beta^* \in \mathcal{S}_\infty := \{\beta \in \mathbb{R}^d \mid Q\beta = Q_Y\}.$$

**Theorem 3.3** (Identifiability for high-dimensional instrument and dense $\beta^*$). *Assume Assumption 1. We have* $\mathcal{S}_\infty = \{\beta^*\}$ *if and only if*

$$\operatorname{rank}(Q) = d.$$

Defining $\mathcal{S}_{0,\infty} := \arg\min_{\beta \in \mathcal{S}_\infty} \|\beta\|_0$, we also prove the result for sparse $\beta^*$.

**Theorem 3.4** (Identifiability for high-dimensional instrument and sparse $\beta^*$). *Assume Assumption 1. Fix $s^* \leqslant d$. The following statements are equivalent:*

*(i) (*Identifiability via sparsest solution*) For all $\beta^* \in \mathbb{R}^d$ with $\|\beta^*\|_0 \leqslant s^*$, it holds that $\mathcal{S}_{0,\infty} = \{\beta^*\}$.*

*(ii) (*Restricted nullspace*) $\ker(Q) \cap \Sigma_{2s^*} = \{0\}$.*

## 4. Estimation

For the setting with finite-dimensional instrument and sparse $\beta^*$ we propose a GMM estimator with $\ell_1$ regularization. Moreover, we demonstrate that in the high-dimensional instrument regime standard two-sample IV estimators are asymptotically biased, and we introduce a cross-moment GMM estimator (for dense and sparse $\beta^*$) that remains consistent and asymptotically normal. As in Section 2.1 we assume that $\{(I_i, Y_i)\}_{i=1}^n$ are i.i.d. and $\{(\tilde{I}_j, \tilde{X}_j)\}_{j=1}^{\tilde{n}}$ are i.i.d., but, additionally, also that the two samples are independent. Define $N := n + \tilde{n}, \tau_n := \frac{n}{N} \to \tau \in (0,1), \tilde{\tau}_n := \frac{\tilde{n}}{N} \to \tilde{\tau} \in (0,1)$.

### 4.1. Finite-Dimensional Instrument

For completeness and to help with intuition, we first state the consistency and asymptotic normality for dense $\beta^*$. Similar results have appeared in the standard two-sample IV literature (Angrist and Krueger, 1991; Inoue and Solon, 2010; Burgess et al., 2013; Bowden et al., 2015).

For a positive definite weighting matrix $W_N \in \mathbb{R}^{m \times m}$, define the sample moment

$$\hat{g}_N(\beta) := \widehat{\operatorname{Cov}}(I, Y) - \widehat{\operatorname{Cov}}(\tilde{I}, \tilde{X})\beta \in \mathbb{R}^m, \qquad (4.1)$$

and the UP-GMM estimator

$$\hat{\beta}_{\mathrm{GMM}}^{\mathrm{UP}}(W_N) \in \arg\min_{\beta \in \mathbb{R}^d} \hat{g}_N(\beta)^\top W_N \hat{g}_N(\beta), \qquad (4.2)$$

---

[1]This setting is natural when considering categorical instruments as it ensures that $\operatorname{Var}(X)$ stays bounded as $m \to \infty$. For details see Example 4.4.

where we use the standard estimators $\widehat{\operatorname{Cov}}(I, Y) :=$ $\frac{1}{n} \sum_{i=1}^n I_i Y_i$ and $\widehat{\operatorname{Cov}}(\tilde{I}, \tilde{X}) := \frac{1}{\tilde{n}} \sum_{j=1}^{\tilde{n}} \tilde{I}_j \tilde{X}_j^\top$. If we choose $W_N = \mathrm{Id}$ then UP-GMM coincides with TS-IV. Define $\Omega := \tau^{-1}\Omega_m + \tilde{\tau}^{-1}\Omega_c(\beta^*), \Omega_m :=$ $\operatorname{Var}(IY), \Omega_c(\beta^*) := \operatorname{Var}(\tilde{I}\tilde{X}^\top\beta^*)$, and denote by $\hat{\Omega}$ the sample version of $\Omega$ (see Appendix K for details). The asymptotic variance of $\hat{\beta}_{\mathrm{GMM}}^{\mathrm{UP}}$ is minimized choosing $W_N$ to be equal $\widehat{W} := \hat{\Omega}^{-1}$ in (4.2). For details, see Appendix K.

**Proposition 4.1** (informal version of Proposition K.1). *Under some technical assumptions and if $\operatorname{rank}(\operatorname{Cov}(I, X)) = d$ we have that*

$$\hat{\beta}_{\mathrm{GMM}}^{\mathrm{UP}}(W_N) \xrightarrow{p} \beta^*. \qquad (4.3)$$

*Furthermore, $\hat{\beta}_{\mathrm{GMM}}^{\mathrm{UP}}(W_N)$ is asymptotically normal.*

**Sparse $\beta^*$ and penalized GMM.** When the full-rank condition in Proposition 4.1 fails, dense identification is in general not possible: there are too few independent moment conditions to recover an arbitrary $d$-dimensional $\beta^*$. We instead impose sparsity and require only that the moment operator be well conditioned on sparse cones (a restricted-eigenvalue/compatibility condition). Then the $\ell_1$-penalized GMM in (4.4) consistently estimates $\beta^*$, attains the rates with $\lambda \asymp \sqrt{1/N}$, and recovers the support under a beta-min condition (see Theorem 4.2). More formally, when $\|\beta^*\|_0 = s^* \leqslant d$, we estimate $\beta^*$ by the $\ell_1$–penalized UP-GMM

$$\hat{\beta}_{\mathrm{GMM}, \ell_1}^{\mathrm{UP}}(W_N) \in \arg\min_{\beta \in \mathbb{R}^d} \frac{1}{2} \left\| W_N^{1/2} \hat{g}_N(\beta) \right\|_2^2 + \lambda_N \|\beta\|_1,$$
$$(4.4)$$

where $\hat{g}_N(\beta)$ is given by (4.1) and $\lambda_N \asymp 1/\sqrt{N}$.

**Assumption 3.** *(i) Assumption 1 holds, the centering convention and boundedness of fourth moments.*

*(ii) There is $W_0$ positive definite such that $W_N \xrightarrow{p} W_0$.*

*(iii) (Restricted eigenvalue.) There exists $\kappa > 0$ such that for all $S \subseteq [d]$ with $|S| = s^*$ and all $\Delta \in \mathbb{R}^d$ with $\|\Delta\|_2 = 1$ and $\|\Delta_{S^c}\|_1 \leqslant 3\|\Delta_S\|_1$,[2] we have $\kappa \leqslant \|\operatorname{Cov}(\tilde{I}, \tilde{X})\Delta\|_{W_0}^2$.*

Assumption 3 (iii) implies Theorem 3.2 (ii) and therefore Assumption 3 (iii) is at least as strong as Theorem 3.2 (ii). Assumption 3 (iii) is a standard assumption in the Lasso literature (see, e.g., Bühlmann and Van De Geer (2011)).

We can now prove consistency and support recovery. Let $S^* := \operatorname{supp}(\beta^*)$ denote the true support of $\beta^*$.

**Theorem 4.2** (Rates for penalized GMM). *For all $\beta^*$ with $\|\beta^*\|_0 = s^*$, under Assumption 3 and $\lambda_N \asymp 1/\sqrt{N}$, the estimator (4.4) satisfies*

$$\|\hat{\beta}_{\mathrm{GMM}, \ell_1}^{\mathrm{UP}}(W_N) - \beta^*\|_2 = O_p\left(\sqrt{s^*/N}\right).$$

---

[2]We define $\|a\|_W := a^T W a$.

*In particular, $\hat{\beta}^{\mathrm{UP}}_{\mathrm{GMM},\ell_1}(W_N) \xrightarrow{p} \beta^*$. If, in addition, there exists $c > 0$ such that for all $i \in S^*$, we have $|\beta_i^*| \geqslant c$ (beta-min condition), then*

$$\widehat{S}(W_N) := \{j \in [d] \mid |\hat{\beta}^{\mathrm{UP}}_{\mathrm{GMM},\ell_1}(W_N)_j| \geqslant c/2\} \xrightarrow{p} S^*.$$

We provide asymptotic confidence intervals for UP-GMM with $\ell_1$-regularization in Appendix E.

### 4.2. High-Dimensional Instrument

We first show that the naive TS-IV estimator is asymptotically biased in the high-dimensional instrument setting. This bias is a consequence of measurement error in the estimate of $\mathrm{Cov}(I,X)$ and appears only in two-sample IV. It is therefore distinct from the endogeneity problem that biases the one-sample IV. The measurement error in high-dimensional unpaired IV is related to the attenuation bias in the paired weak-instrument setting (Angrist and Krueger, 1995; Choi et al., 2018). In the weak-instrument literature attenuation appears as a result of removing the weak-instrument bias with the split-sample IV estimator which as a consequence introduces attenuation. We show that high-dimensional instrument unpaired IV suffer from a similar bias and prove that sample-splitting is an effective solution for the measurement-error bias.

**Classical estimators are biased in paired and unpaired settings.** Assume Assumption 1. For notational simplicity we consider $\tilde{n} = n = rm$ (for some $r \in \mathbb{N}$), $d = 1$ and $m \to \infty$. We show that the naive two-sample IV estimator is asymptotically biased for general $d \in \mathbb{N}$, $r = n/m \geqslant 1$ and $\tilde{r} = \tilde{n}/m \geqslant 1$. The naive estimator solves (4.2) with $W_N = Id$, that is,

$$
\hat{\beta} := \frac{\widehat{\mathrm{Cov}}(\tilde{I},\tilde{X})^\top \widehat{\mathrm{Cov}}(I,Y)}{\widehat{\mathrm{Cov}}(\tilde{I},\tilde{X})^\top \widehat{\mathrm{Cov}}(\tilde{I},\tilde{X})}
$$

$$
= \beta^* \frac{\widehat{\mathrm{Cov}}(\tilde{I},\tilde{X})^\top \widehat{\mathrm{Cov}}(I,X)}{\widehat{\mathrm{Cov}}(\tilde{I},\tilde{X})^\top \widehat{\mathrm{Cov}}(\tilde{I},\tilde{X})} + \frac{\widehat{\mathrm{Cov}}(\tilde{I},\tilde{X})^\top \widehat{\mathrm{Cov}}(I,\varepsilon)}{\widehat{\mathrm{Cov}}(\tilde{I},\tilde{X})^\top \widehat{\mathrm{Cov}}(\tilde{I},\tilde{X})}.
$$

If $m$ were constant and $n \to \infty$, then $\widehat{\mathrm{Cov}}(\tilde{I},\tilde{X}) \xrightarrow{p} \mathrm{Cov}(I,X)$, $\widehat{\mathrm{Cov}}(I,\varepsilon) \xrightarrow{p} 0$ and therefore $\hat{\beta} \xrightarrow{p} \beta^*$. However, in the high-dimensional case where $\tilde{n}/m \to \tilde{r} \in (0,\infty)$ in general it does not hold that $\|\widehat{\mathrm{Cov}}(\tilde{I},\tilde{X}) - \mathrm{Cov}(I,X)\|_2^2 \xrightarrow{p} 0$ resulting in an inconsistent estimator.

**Lemma 4.3.** *Assume Assumption 4 (see below) with $d = 1$ and $\beta^* \neq 0$. Additionally, assume that $\mathrm{tr}(\Sigma_{IX}) \to b \in (0,\infty)$ and $\sup_m \mathbb{E}[\|IX - c\|_2^4] \leqslant \infty$. Then*

$$\hat{\beta} \xrightarrow{p} \beta^* Q/(Q + b/\tilde{r}) \neq \beta^*.$$

In another setting, in the paired high-dimensional instrument setting, 2SLS is known to be biased, too, but for a

different reason: here, $\widehat{\mathrm{Cov}}(I,\varepsilon) \not\to 0$. This is often called many-instrument bias and methods such as LIML (Anderson and Rubin, 1949; Fuller, 1977) or SS-IV (Angrist and Krueger, 1995) have been developed to remove the bias.

A simple solution to the measurement-error problem in unpaired IV is to use sample splitting in the denominator, i.e., we divide the $(\tilde{I},\tilde{X})$-data into two samples $A \subseteq [\tilde{n}]$ and $B = A^c$ of equal size. Then define $E := \widehat{\mathrm{Cov}}(\tilde{I},\tilde{X}) - \mathrm{Cov}(I,X)$, $E_A := \widehat{\mathrm{Cov}}_A(\tilde{I},\tilde{X}) - \mathrm{Cov}(I,X)$ and $E_B := \widehat{\mathrm{Cov}}_B(\tilde{I},\tilde{X}) - \mathrm{Cov}(I,X)$ (where $\widehat{\mathrm{Cov}}_A(\tilde{I},\tilde{X})$ is estimated on fold $A$ and $\widehat{\mathrm{Cov}}_B(\tilde{I},\tilde{X})$ on fold $B$). Because $E_A \perp\!\!\!\perp E_B$ we get $E_A^\top E_B \xrightarrow{p} 0$ recovering consistency in the high-dimensional setting. We now make this idea precise.

**Unbiased estimators via cross moments.** Split the $(\tilde{I},\tilde{X})$-sample into $K \geqslant 2$ disjoint, equal-sized folds and let $\widehat{\mathrm{Cov}}_k(\tilde{I},\tilde{X}), k \in [K]$, be the foldwise covariance vectors (computed only within fold $k$). Define

$$C_{XX} := \frac{m}{K(K-1)} \sum_{k \neq h} \widehat{\mathrm{Cov}}_h(\tilde{I},\tilde{X})^\top \widehat{\mathrm{Cov}}_k(\tilde{I},\tilde{X}),$$

$$C_{XY} := m\widehat{\mathrm{Cov}}(\tilde{I},\tilde{X})^\top \widehat{\mathrm{Cov}}(I,Y).$$

and $\hat{g}(\beta) := C_{XY} - C_{XX}\beta$. Define the SPLITUP estimator

$$\hat{\beta}^{\mathrm{SPLITUP}}(W_N) \in \arg\min_\beta \hat{g}(\beta)^\top W_N^{-1} \hat{g}(\beta),$$

where $W_N$ is a weight matrix such that $W_N \to W_0$ as $m \to \infty$ and where $W_0$ is positive definite.

**Assumption 4.** *Assume $d \in \mathbb{N}$ and $K \geqslant 2$ are fixed, and $n/m \to r$, $\tilde{n}/m \to \tilde{r}$ as $m \to \infty$. Additionally, assume all of the following.*

*(i) Assumption 1 holds and we use the centering convention.*

*(ii) The limit $Q := \lim_{m \to \infty} m \mathrm{Cov}(\tilde{I},\tilde{X})^\top \mathrm{Cov}(\tilde{I},\tilde{X}) \in \mathbb{R}^{d \times d}$ exists and is positive definite.*

*(iii) Writing $\Sigma_{IX} := \mathrm{Var}(\mathrm{vec}(\tilde{I}\tilde{X})) \in \mathbb{R}^{md \times md}$ and $\Sigma_{IY} := \mathrm{Var}(IY) \in \mathbb{R}^{m \times m}$, there exists $C < \infty$ such that uniformly for all $m \in \mathbb{N}$ such that $m\|\Sigma_{IX}\|_{\mathrm{op}} \leqslant C$ and $m\|\Sigma_{IY}\|_{\mathrm{op}} \leqslant C$.*

Assumption 4 (ii) replaces (3.2) from the low-dimensional case. Assumption 4 (iii) imposes a uniform bound on the operator norms of the covariance of $IX$ and $IY$, meaning that the cross–sectional noise in the first stage and reduced form does not explode as the number of instruments grows.

**Example 4.4** (categorical instruments). *Let $K \sim \mathrm{Unif}\{1,\ldots,m\}$, $\bar{I} = e_K$, and $I = \bar{I} - \mathbb{E}[\bar{I}] = e_K - \frac{1}{m}\mathbb{1}$. Consider a categorical first stage $X = \mu^\top \bar{I} + \varepsilon = \mu_K + \varepsilon, \mathbb{E}[\varepsilon \mid K] = 0$, and assume uniformly bounded 8th moments and $\mathrm{Var}(\mu^\top \bar{I}) = \frac{1}{m}\|\mu - \bar{\mu}\mathbb{1}\|_2^2 \to Q \in (0,\infty)$.*

Then $\mathrm{Cov}(I, X) = \mathbb{E}[IX] = \frac{1}{m}(\mu - \bar{\mu}\mathbb{1})$ *and*

$$m\|\mathrm{Cov}(I, X)\|_2^2 = \frac{1}{m}\|\mu - \bar{\mu}\mathbb{1}\|_2^2 \to Q,$$

*so Assumption 4 (ii) holds (here $d = 1$). For Assumption 4 (iii), write $Z = IX \in \mathbb{R}^m$. For each coordinate $i$, $Z_i = (\mathbb{1}\{K = i\} - 1/m)X$, so $\mathrm{Var}(Z_i) = O(1/m)$ and for $i \neq j$ we have $|\mathrm{Cov}(Z_i, Z_j)| = O(1/m^2)$. Therefore each row sum of $\Sigma_{IX} = \mathrm{Var}(IX)$ is $O(1/m)$, implying $\|\Sigma_{IX}\|_{\mathrm{op}} \leqslant \|\Sigma_{IX}\|_1 = O(1/m)$, hence $m\|\Sigma_{IX}\|_{\mathrm{op}} = O(1)$ uniformly in $m$. The same argument applies to $\Sigma_{IY} = \mathrm{Var}(IY)$ (under bounded second moments of $Y$), giving $m\|\Sigma_{IY}\|_{\mathrm{op}} = O(1)$. Finally, $\mathrm{tr}(\Sigma_{IX}) = \sum_{i=1}^m \mathrm{Var}(Z_i) \to b \in (0, \infty)$ provided $\mathbb{E}[X^2]$ stays bounded away from $0$.*

**Theorem 4.5** (Consistency). *Assume Assumption 4. Let $d \in \mathbb{N}$ and $K \geqslant 2$ be fixed, and $n/m \to r \in (0, \infty)$, $\tilde{n}/m \to \tilde{r} \in (0, \infty)$. Let $W_N \in \mathbb{R}^{d \times d}$ be a sequence of positive definite weight matrices such that $W_N \to W_0$ in probability with $W_0 \succ 0$. Then*

$$\hat{\beta}^{\textsc{SplitUP}}(W_N) \xrightarrow{p} \beta^*.$$

### 4.3. High-dimensional instrument with sparse $\beta^*$

We assume that $n = rm$ and $\tilde{n} = \tilde{r}m$ for the constants $r, \tilde{r} \in (0, \infty)$, while the causal effect is sparse with $s^* := \|\beta^*\|_0 \ll d$. Define the $\ell_1$–penalized estimator

$$\hat{\beta}_{\ell_1}^{\textsc{SplitUP}}(W_m) \in \arg\min_{\beta \in \mathbb{R}^d} \frac{1}{2}\big\|W_m^{1/2}\big(C_{XY} - C_{XX}\beta\big)\big\|_2^2 + \lambda_m\|\beta\|_1.$$

We take $\lambda_m \asymp 1/\sqrt{m}$.

**Assumption 5.** *Assume all of the following.*

*(i) Assumption 1, the centering convention and with bounded fourth moments.*
*(ii) The limit $Q := \lim_{m\to\infty} m\,\mathrm{Cov}(\tilde{I}, \tilde{X})^\top \mathrm{Cov}(\tilde{I}, \tilde{X})$ exists and is well-defined.*
*(iii) There is $W_0$ positive definite such that $W_m \xrightarrow{p} W_0$.*
*(iv) There exists $\kappa > 0$ such that for all supports $S \subset [d]$ with $|S| = s^*$ and all $\Delta \in \mathbb{R}^d$ with $\|\Delta\|_2 = 1$ and $\|\Delta_{S^c}\|_1 \leqslant 3\|\Delta_S\|_1$ we have $\kappa \leqslant \|Q\Delta\|_{W_0}^2$.*
*(v) Assumption 4 (iii) holds.*

Assumption 5 (iv) is a standard assumption in the Lasso literature (see, e.g., Bühlmann and Van De Geer (2011)).

**Theorem 4.6.** *Let $S^* := \mathrm{supp}(\beta^*)$ with $|S^*| = s^*$. Under Assumption 5 and $\lambda_m \asymp \sqrt{1/m}$, the estimator $\hat{\beta}_{\ell_1}^{\textsc{SplitUP}}$ satisfies*

$$\|\hat{\beta}_{\ell_1}^{\textsc{SplitUP}}(W_m) - \beta^*\|_2 = O_p\Big(\sqrt{\frac{s^*}{m}}\Big).$$

*If, in addition, there exists $c > 0$ such that $\min_{j \in S^*} |\beta_j^*| \geqslant c$ (beta–min), then*

$$\hat{S}(W_m) := \{j \in [d] \mid |\hat{\beta}_{\ell_1}^{\textsc{SplitUP}}(W_m)|_j \geqslant c/2\} \xrightarrow{p} S^*.$$

In practice we can reduce the variance of the cross-fitting estimator by split averaging and categorical-instrument simplification. A natural question is whether the split-average admits a closed form when we average over all possible splits. The answer is "yes":

$$\lim_{H\to\infty} \frac{1}{H} \sum_{h=1}^{H} C_{XX}(A_h, B_h)$$
$$= \frac{\tilde{n}}{\tilde{n}-1}\|\widehat{\mathrm{Cov}}(\tilde{I}, \tilde{X})\|_F^2 - \frac{1}{\tilde{n}(\tilde{n}-1)}\sum_{i=1}^{\tilde{n}}\|\tilde{I}_i\tilde{X}_i^\top\|_F^2.$$

Thus, the $H \to \infty$ split-average equals the usual plug-in quadratic form $\widehat{\mathrm{Cov}}(\tilde{I}, \tilde{X})^\top\widehat{\mathrm{Cov}}(\tilde{I}, \tilde{X})$ minus a diagonal 'self-inner-product' correction. We provide details in Appendix F.

## 5. Experiments

We compare SplitUP (with identity weighting) and UP-GMM against standard two-sample baselines (TS-IV and TS-2SLS) on synthetic and real-world data. Implementation details, hyperparameters, and data-generation specifics are deferred to Appendix H, additional experiments to Appendix I. All code required to replicate our experiments can be found here: https://github.com/fschur/splitup-iv.

### 5.1. Synthetic Experiments

We study three regimes with categorical instruments (the corresponding experiments with continuous instruments are reported in Appendix I): finite-dimensional instruments with sparse $\beta^*$ (Setting 1), high-dimensional instruments with dense $\beta^*$ (Setting 2), and high-dimensional instruments with sparse $\beta^*$ (Setting 3). Throughout, we use balanced categories and balanced sample sizes ($\tilde{n} = n$). We also report results as a function of the sample-to-instrument ratio $\frac{n}{m}$ which matches the high-dimensional scaling used in Section 3.2. Furthermore, TS-IV and TS-2SLS are numerically equivalent for the categorical setting with balanced samples sizes. We therefore report TS-2SLS only in the continuous settings. For all experiments, shaded regions indicate confidence intervals around the mean computed over 50 independent runs.

For categorical instruments, we exploit the simplifications described in Appendix F. In particular, for SplitUP we consider (a) Monte Carlo averaging over random splits (with $H$ splits) and (b) the closed-form infinite-split version, denoted SplitUP (analytic). As shown in Figure 11,

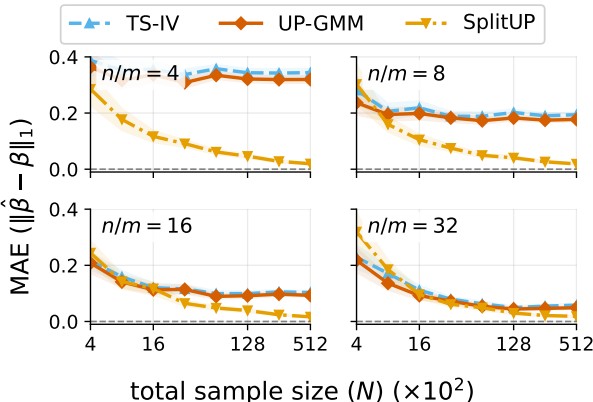

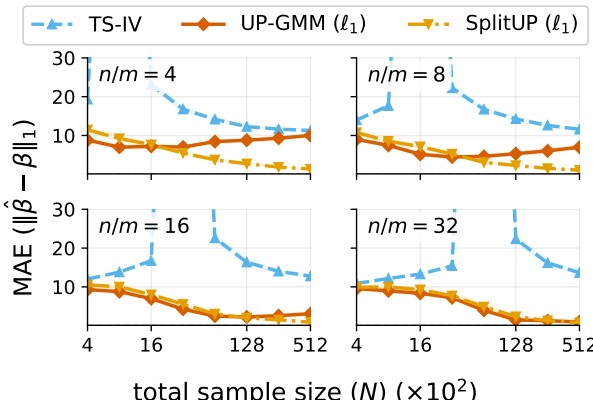

*Figure 3.* **Setting 2 (high-dimensional instruments, dense $\beta^*$).** In the high-dimensional instrument regime, TS-IV and UP-GMM (and TS-2SLS) are asymptotically biased. The cross-moment denominator in SPLITUP removes this bias and is the only method that is consistent in this setting. Consistent with theory, the bias of the naive estimators decreases as $n/m$ increases, i.e., as the problem becomes less high-dimensional.

SPLITUP with $H = 10$ and SPLITUP (analytic) are numerically indistinguishable in our setting; we therefore report SPLITUP (analytic) in all remaining plots.

**Setting 1: Finite-dimensional instruments, sparse $\beta^*$.** Results are shown Appendix I.1 (Figure 10). Since the moment matrix is rank deficient (even asymptotically) the unregularized TS-IV estimator is inconsistent. Both UP-GMM($\ell_1$) and SPLITUP($\ell_1$) exploit sparsity and therefore recover the true causal effect (UP-GMM($\ell_1$) performs slightly better as the bias correction is not necessary here).

**Setting 2: High-dimensional instruments, dense $\beta^*$.** Results are shown in Figure 3. In the high-dimensional instrument regime, the naive plug-in denominator induces a persistent measurement-error bias in two-sample IV, making both TS-IV and UP-GMM asymptotically biased. The cross-moment construction in SPLITUP removes this bias and is the only method that remains consistent in this setting. As predicted by theory, the bias of the naive estimators decreases as $n/m$ increases, i.e., as the problem becomes less high-dimensional.

**Setting 3: High-dimensional instruments, sparse $\beta^*$.** Results are shown in Figure 4. This setting combines the challenges from Settings 1 and 2: identification relies on sparsity, and high-dimensional instruments induce measurement-error bias for plug-in denominators. Consequently, TS-IV suffers from both effects, while UP-GMM addresses the sparse identification aspect but still inherits the high-dimensional measurement-error bias. By combining sparsity with a cross-fit denominator, SPLITUP

*Figure 4.* **Setting 3 (high-dimensional instruments, sparse $\beta^*$).** This setting combines sparse identification with high-dimensional measurement-error bias from plug-in denominators. As a result, TS-IV suffers from both effects, while UP-GMM addresses the sparsity aspect but still inherits the high-dimensional bias. By combining sparsity with a cross-fit denominator, SPLITUP is consistent and achieves the smallest error as $n$ grows. We also observe a transient peaking phenomenon for TS-IV. We hypothesize that it is due to small but non-zero eigenvalues of the noise matrix. See Appendix J for details.

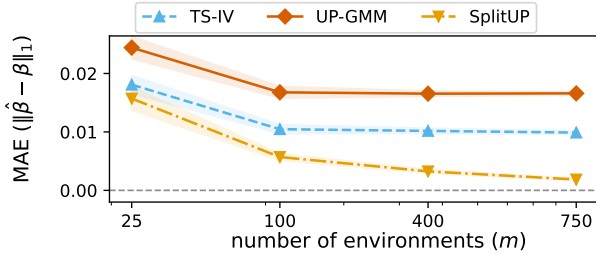

*Figure 5.* **Causal chambers light tunnel.** For each environment we generate $n/m = 8$ observations and let $m$ vary. The graphs suggest that, as predicted by theory, TS-IV and UP-GMM are asymptotically biased while SPLITUP is consistent. Confidence intervals are computed based on 50 resamples.

is consistent and empirically achieves the smallest error as $n$ grows. SPLITUP($\ell_1$) outperforms TS-IV and UP-GMM($\ell_1$) even in settings with $d > m$, i.e., when the causal effect is high-dimensional in addition to the instruments. The results are reported in Appendix I.4.

### 5.2. Experiment: Causal Chambers Light Tunnel

We evaluate our algorithms on data generated by the light-tunnel causal chamber of Gamella et al. (2025), which allows for automated data collection with programmable actuators and photodiode sensors. We use the red-channel intensity (red) as an unobserved confounder $U$ (sampled uniformly from $0, \dots, 255$) that causes the first and second infrared photodiode readings, $X = \text{ir\_1}$ (treatment) and $Y = \text{ir\_2}$ (outcome). To identify the causal effect of $X$

on $Y$ (which we artificially create, using $M := 0.0625 \cdot X$ with $M$ being another LED `l_21`) , we use the upstream infrared LED $I = $ `led_1_ir` (sampled uniformly with replacement from $m$ discrete levels in $(0, \ldots, 749)$ as an instrument that shifts $X$ while remaining independent of $U$ and affecting $Y$ only through $X$ by design of the protocol. We sample $r = n/m$ observations per value of $I$. By construction, this dataset is paired. We artificially unpair it by removing either $X$ or $Y$ in each data point. Although $U$ is treated as unobserved by the estimator, it is logged by the experimenter, enabling oracle targets via covariate adjustment on the paired dataset. We provide further details in Appendix H.4.

We consider $r = n/m = 8$ observations per environment. The results are shown in Figure 5. As predicted by theory, SPLITUP seems consistent and the biases of TS-IV and UP-GMM do not seem to vanish. In finite samples, UP-GMM can underperform TS-IV because it estimates and inverts an (asymptotically) optimal weighting matrix. This seems to be the case here since when we replace this weight by the identity, UP-GMM performs similarly to TS-IV, suggesting that the gap is driven by weighting-matrix instability rather than by the moment condition itself.

## 6. Discussion and Future Work

We studied causal effect estimation from unpaired data under hidden confounding, focusing on regimes with many environments, few repetitions per environment, and potentially sparse causal effects. Our results show that consistent estimation is possible in this setting, but they rely on structural assumptions, notably cross-sample moment invariance and exclusion-type conditions. Important directions for future work, particularly for broadening applicability in practice, include uncertainty quantification in high-dimensional settings and extending the framework beyond linear structural effects for example via fixed feature transforms.

## Acknowledgment

During part of this work, Felix Schur was supported by a travel grant from G-Research. We thank Juan Gamella for supporting us with the real-world experiments.

## Impact Statement

This paper presents work whose goal is to advance the field of machine learning. There are many potential societal consequences of our work, none of which we feel must be specifically highlighted here.

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

# A. Extended Related Work

**Two–sample IV and two-sample Mendelian randomization.** Methods from the two–sample IV and two-sample (summary-data) Mendelian randomization literature can be applied to our setting (even if they may not be consistent, see Section 4.2). Two-sample IV estimators combine instrument–exposure and instrument–outcome information from separate samples (Angrist and Krueger, 1995; Inoue and Solon, 2010; Pacini and Windmeijer, 2016; Zhao et al., 2019); in econometrics, such two-sample moment combinations have been used to study compulsory schooling and returns to education, for example in age-at-entry and split-sample IV designs (Angrist and Krueger, 1992). Analogous designs underpin two–sample Mendelian randomization (MR) in genetics, where variants act as instruments (Hartwig et al., 2016; Zhao et al., 2020). Beyond these, two–sample MR theory highlights design issues such as winner's-curse and sample overlap, which can bias naive estimators (Pierce and Burgess, 2013; Burgess et al., 2016), as well as pervasive horizontal pleiotropy addressed by mode-based and mixture/likelihood approaches (Hartwig et al., 2017; Qi and Chatterjee, 2019; Morrison et al., 2020). Our unpaired-sample formulation differs from previous work on two-sample IV and Mendelian randomization in that we allow for high-dimensional instruments and/or large number dimension of the treatment. More concretely, we propose consistent estimators in regimes where the ratio of the sample size $n$ to the instrument dimension $m$ converges to a positive constant and/or the dimension of the treatment $d$ is larger than the dimension of the instrument $m$.

**Sparse IV regression.** A growing literature studies instrumental variable models in which the structural coefficient vector $\beta^*$ is sparse and the number of endogenous regressors and instruments can be large. One approach keeps the two-stage structure of 2SLS while imposing an $\ell_1$-penalty in the structural equation. Zhu (2018) analyzes $\ell_1$-regularized 2SLS for triangular models, allowing both endogenous regressors and instruments to exceed the sample size, and establishes high-dimensional consistency and error bounds when the first-stage and structural parameters are sufficiently sparse. A related two-stage regularization framework is developed by Lin et al. (2015), who use sparsity-inducing penalties in both stages to estimate fitted regressors and a sparse structural equation. In large systems of simultaneous equations, Chen et al. (2018) propose two-stage penalized least squares with ridge in the first stage and adaptive Lasso in the second, proving oracle-type results for support recovery and asymptotic normality of the selected nonzero effects. For inference in these settings, Gold et al. (2020) construct a desparsified (one-step) GMM update for high-dimensional IV and show asymptotic normality of the debiased estimator, yielding valid confidence intervals for components of $\beta^*$ when initialized by two-stage Lasso-type estimators. Complementing these results, Belloni et al. (2022) consider models with many endogenous variables and many instruments, using orthogonalized moments and Lasso only for nuisance estimation; they provide uniformly valid post-selection inference via multiplier bootstrap. More recently, Pfister and Peters (2022) establish identifiability of a sparse $\beta^*$ in linear IV models—which may be achievable when the instrument dimension is smaller than $d$—derive graphical conditions, and propose the SPACEIV estimator (paired observations). Tang et al. (2023) also adopt the sparse causation premise and construct a synthetic instrument from $X$; under linear SEMs they show identifiability despite unmeasured confounding and cast estimation as $\ell_0$-penalization (paired observations). Complementing these, Huang et al. (2024) study sparse causal effects with two-sample summary statistics (two-sample MR) when the variance of the instrument $\mathrm{Var}(I)$ is invertible, adapting SPACEIV to a summary-data regime. However, they do not provide theoretical guarantees such as consistency or asymptotic normality for their estimator. A second strand regularizes IV estimation directly through penalized moment conditions, without an explicit two-stage Lasso second stage. Caner (2009) introduces a Lasso-type GMM estimator that adds an $\ell_1$-style penalty to the GMM criterion and derives model selection and estimation guarantees under sparsity. Building on this idea, Shi (2016) studies GMM-Lasso for linear structural models with many endogenous regressors and proves oracle inequalities for sparse $\beta^*$. In a related but distinct formulation, Fan and Liao (2014) propose penalized focused GMM (FGMM) to address incidental endogeneity in high dimensions, establishing support recovery and asymptotic normality for the active coefficients under suitable conditions.

**Many-instrument and weak-instruments.** A large literature studies weak and many instruments in standard single-sample IV. We also consider the many instrument setting (or high-dimensional instrument) setting. Since we assume that the norm of the instrument is constant with increasing sample size (which implies that the magnitude of each component of the instrument is decreasing in the sample size) our work also relates to the weak instrument literature. Early work by Staiger and Stock (1997) showed that results in conventional asymptotics can be misleading when instruments are weak, motivating alternative limiting frameworks and bias approximations. Many-instrument asymptotics (in the sense of the work by Bekker (1994) and the subsequent contributions such as the ones by Donald and Newey (2001) and Hansen et al. (2008)) analyze the behavior of IV estimators when the number of instruments grows with the sample size, and motivate procedures that remain well behaved in that regime. Practical diagnostics and test procedures for weak instruments are

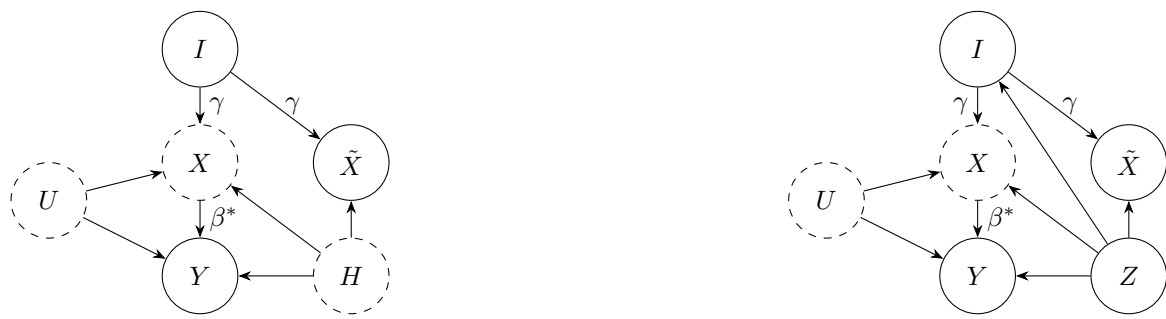

*Figure 6.* Various causal graphs induced by SCMs. Although the shown graphs deviate from the standard two-sample IV setting identification (under appropriate rank conditions) is still possible within our setup.

surveyed by Andrews et al. (2019), who synthesize theory and practice for both weak and many instruments. Stock and Yogo (2005) develop widely used critical values for weak-IV tests based on first-stage $F$-statistics. High-dimensional extensions that allow for many controls and potential instruments include the post-double-selection methods of Belloni et al. (2014), which provide uniformly valid inference on treatment effects after model selection in sparse high-dimensional regressions, and related work on $\ell_1$-regularized IV and debiased machine learning for structural parameters (Belloni et al., 2012). Choi et al. (2018) develop weak-instrument robust inference for two-sample IV, constructing Anderson–Rubin, Kleibergen–$K$, and conditional likelihood ratio-type tests whose validity does not rely on strong instruments and that remain robust to heterogeneous second moments across samples. We analyze the many-instrument regime for unpaired observations, a setting that appears to be largely unexplored in the existing literature. We show that naively constructed estimators in this regime are asymptotically biased due to a measurement-error effect that is specific to the two-sample, high-dimensional instrument (i.e., many-instrument) setting and does not arise in standard paired-sample estimators. We show how to remove this bias and obtain consistent, asymptotically normal estimators.

Going beyond existing literature, we characterize identifiability for unpaired IV estimators in the regime $d > m$ (with a sparse structural parameter) and establish consistency and asymptotic normality of straightforward estimators. Additionally, we characterize identifiability when the instrument is high-dimensional ($m \to \infty$). For this setting, we propose novel estimators and prove consistency.

## B. Examples of Distributions Satisfying Assumption 1

Assumption 1 (and (2.1)) is rather general: many distributions satisfy Assumption 1 while deviating substantially from the standard two-sample IV setup. In this section, we assume that the data are generated by structural causal models (SCMs) whose induced graphs are shown in Figure 6. The parameter of interest is $\beta^*$. We assume that the instrument $I$ affects $X$ and $\tilde{X}$ in the same way up to second moments, which we depict by a shared coefficient $\gamma$.

**Example 1.** In this example there is no $\tilde{Y}$ variable. The variables $X$, $Y$, and $\tilde{X}$ are confounded by an unobserved variable, and we may have $I = \tilde{I}$. If the distribution is generated by an SCM with induced graph given by Figure 6 (left), then Assumption 1 is satisfied. Under the usual additional conditions (a rank condition in the dense setting and a restricted nullspace condition in the sparse setting), $\beta^*$ remains identifiable.

**Example 2.** In addition, there may be an observed confounder $Z$ that affects $X$, $\tilde{X}$, and $Y$; see Figure 6 (middle). In this case, Assumption 1 holds conditional on $Z$, i.e., $\text{Cov}(I, Y \mid Z) = \beta^* \text{Cov}(I, \tilde{X} \mid Z)$. With the same additional conditions as above (rank in the dense case; restricted nullspace in the sparse case), identification again follows.

## C. Discussion of Assumption 1

### C.1. Assumption 1 is Necessary

Theorem C.1 shows that in the non-degenerate, scalar case Assumption 1 is a necessary conditions for identifiability.

**Theorem C.1.** *Consider the scalar case $m = d = 1$. Let $P_{IY}$ and $P_{\tilde{I}\tilde{X}}$ be centered distributions of $(I, Y)$ and $(\tilde{I}, \tilde{X})$*

*with finite second moments and* $\mathrm{Var}(I) > 0$. *Define*

$$b := \mathrm{Cov}(\tilde{I}, \tilde{X}), \qquad c := \mathrm{Cov}(I, Y), \qquad g(I) := \mathbb{E}[Y \mid I].$$

*Let* $\mathcal{M}(P_{IY}, P_{\tilde{I}\tilde{X}})$ *be the class of all joint distributions of*

$$(I, X, Y, \tilde{I}, \tilde{X})$$

*such that*

(a) *the* $(I, Y)$*-marginal is* $P_{IY}$,
(b) *the* $(\tilde{I}, \tilde{X})$*-marginal is* $P_{\tilde{I}\tilde{X}}$,
(c) *there exist* $\beta^* \in \mathbb{R}$ *and a square-integrable random variable* $\varepsilon$ *such that*

$$Y = \beta^* X + \varepsilon \qquad almost \ surely.$$

*Then:*

(i) *If* $(c, b) \neq (0, 0)$, *then there exist two models in* $\mathcal{M}(P_{IY}, P_{\tilde{I}\tilde{X}})$ *with different causal effects* $\beta_1^* \neq \beta_2^*$ *that induce the same observed distributions of* $(I, Y)$ *and* $(\tilde{I}, \tilde{X})$, *both satisfy Assumption 1(ii), both violate Assumption 1(i), and both satisfy*

$$\mathrm{Cov}(X, \varepsilon) \neq 0.$$

(ii) *If it is not the case that* $b = 0$ *and* $g(I) = 0$ *almost surely, then there exist two models in* $\mathcal{M}(P_{IY}, P_{\tilde{I}\tilde{X}})$ *with different causal effects* $\beta_1^* \neq \beta_2^*$ *that induce the same observed distributions of* $(I, Y)$ *and* $(\tilde{I}, \tilde{X})$, *both satisfy Assumption 1(i), both violate Assumption 1(ii), and both satisfy*

$$\mathrm{Cov}(X, \varepsilon) \neq 0.$$

*Proof.* We construct observationally equivalent completions of the same observed law.

Fix an auxiliary random variable $V$ such that

$$V \perp\!\!\!\perp (I, Y, \tilde{I}, \tilde{X}), \qquad \mathbb{E}[V] = 0, \qquad \mathrm{Var}(V) = \sigma^2 > 0.$$

We may always enlarge the probability space so that such a $V$ exists.

**Proof of Item (i).** Assume $(c, b) \neq (0, 0)$. Choose two distinct nonzero numbers $\beta_1^*, \beta_2^*$ such that

$$\frac{c}{\beta_j^*} \neq b, \qquad j = 1, 2.$$

This is possible because: if $c = 0$, then $b \neq 0$ and any two distinct nonzero $\beta_j^*$ work; if $c \neq 0$, there is at most one forbidden value $\beta = c/b$ (when $b \neq 0$).

For $j \in \{1, 2\}$ define

$$X_j := \frac{g(I)}{\beta_j^*} + V, \qquad \varepsilon_j := Y - \beta_j^* X_j.$$

Then

$$Y = \beta_j^* X_j + \varepsilon_j$$

almost surely, so each construction belongs to the class $\mathcal{M}(P_{IY}, P_{\tilde{I}\tilde{X}})$. Moreover, the observed marginals of $(I, Y)$ and $(\tilde{I}, \tilde{X})$ are unchanged, since only the latent variable $X$ has been modified.

Next,

$$\mathbb{E}[\varepsilon_j \mid I] = \mathbb{E}[Y \mid I] - \beta_j^* \mathbb{E}[X_j \mid I] = g(I) - \beta_j^* \left( \frac{g(I)}{\beta_j^*} + \mathbb{E}[V \mid I] \right) = 0,$$

because $\mathbb{E}[V \mid I] = 0$. Hence Assumption 1(ii) holds.

Since all variables are centered,

$$\mathrm{Cov}(I, X_j) = \frac{1}{\beta_j^*}\,\mathrm{Cov}(I, g(I)) = \frac{1}{\beta_j^*}\,\mathrm{Cov}(I, Y) = \frac{c}{\beta_j^*} \neq b = \mathrm{Cov}(\tilde{I}, \tilde{X}),$$

so Assumption 1(i) fails.

Finally, writing $R := Y - g(I)$, we have $\mathbb{E}[R \mid I] = 0$ and

$$\varepsilon_j = Y - \beta_j^*\Big(\frac{g(I)}{\beta_j^*} + V\Big) = R - \beta_j^* V.$$

Therefore

$$\mathrm{Cov}(X_j, \varepsilon_j) = \mathrm{Cov}\Big(\frac{g(I)}{\beta_j^*} + V,\ R - \beta_j^* V\Big) = -\beta_j^*\,\mathrm{Var}(V) \neq 0,$$

because the mixed terms vanish by independence and the tower property. Thus both models are nontrivial in the sense that $\varepsilon_j$ is correlated with $X_j$.

Since $\beta_1^* \neq \beta_2^*$, this proves Item (i).

**Proof of Item (ii).** Let

$$a := \frac{b}{\mathrm{Var}(I)}.$$

Define

$$X := aI + V.$$

Then

$$\mathrm{Cov}(I, X) = a\,\mathrm{Var}(I) = b = \mathrm{Cov}(\tilde{I}, \tilde{X}),$$

so Assumption 1(i) holds.

We now choose $\beta_1^*, \beta_2^*$ so that Assumption 1(ii) fails and $\mathrm{Cov}(X, \varepsilon) \neq 0$.

If $b = 0$, then $a = 0$ and by assumption $g(I) \neq 0$ with positive probability. In this case choose any two distinct nonzero $\beta_1^*, \beta_2^*$.

If $b \neq 0$, then $a \neq 0$ and the set

$$A_{\mathrm{exo}} := \big\{\beta \in \mathbb{R} : g(I) = \beta a I \text{ almost surely}\big\}$$

contains at most one element. Also the set

$$A_{\mathrm{conf}} := \Big\{\frac{ac}{a^2\,\mathrm{Var}(I) + \sigma^2}\Big\}$$

contains at most one element. Hence we can choose two distinct numbers

$$\beta_1^*, \beta_2^* \notin A_{\mathrm{exo}} \cup A_{\mathrm{conf}}.$$

For $j \in \{1, 2\}$ define

$$\varepsilon_j := Y - \beta_j^* X.$$

Then

$$Y = \beta_j^* X + \varepsilon_j$$

almost surely, so again both constructions belong to $\mathcal{M}(P_{IY}, P_{\tilde{I}\tilde{X}})$ and induce the same observed distributions.

Moreover,

$$\mathbb{E}[\varepsilon_j \mid I] = \mathbb{E}[Y \mid I] - \beta_j^* \mathbb{E}[X \mid I] = g(I) - \beta_j^* a I.$$

By construction of $\beta_j^*$, this is not almost surely zero, so Assumption 1(ii) fails.

Finally,

$$\mathrm{Cov}(X, \varepsilon_j) = \mathrm{Cov}(aI + V,\ Y - \beta_j^*(aI + V)) = ac - \beta_j^*(a^2\,\mathrm{Var}(I) + \sigma^2),$$

which is nonzero by the choice $\beta_j^* \notin A_{\mathrm{conf}}$.

Thus we have constructed two observationally equivalent models with different causal effects $\beta_1^* \neq \beta_2^*$, both satisfying Assumption 1(i), both violating Assumption 1(ii), and both having $\mathrm{Cov}(X, \varepsilon_j) \neq 0$. This proves Item (ii). $\qquad\square$

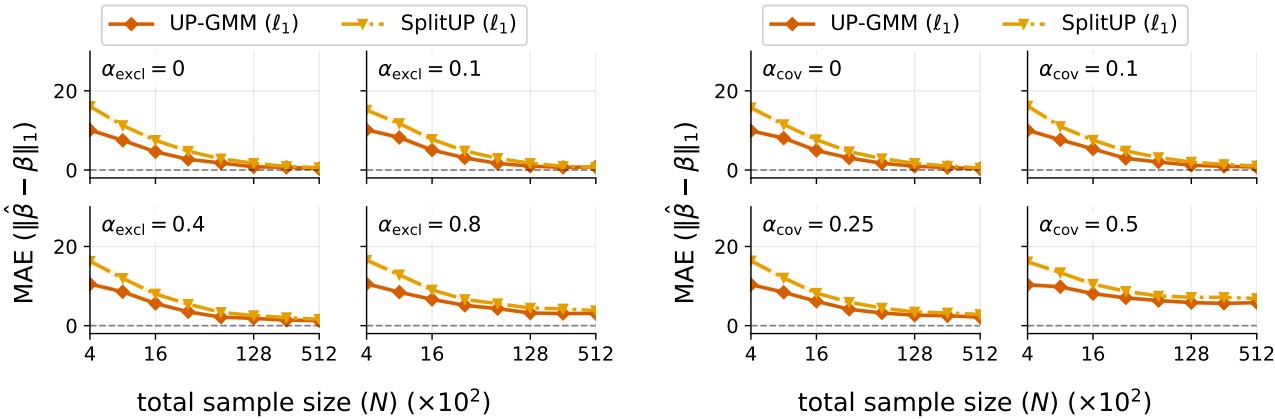

*Figure 7.* **Empirical sensitivity to violations of Assumption 1.** Left: exclusion-restriction violation, where an environment-specific direct effect is added to $Y$. Right: cross-sample covariance mismatch, where the first-stage means are shifted in the $\tilde{X}$-sample. At violation strength zero, Assumption 1 holds. Larger values induce stronger violations and lead to larger estimation error.

### C.2. Empirical Sensitivity Analysis for Violations of Assumption 1

Starting from the sparse categorical Setting 1, where Assumption 1 holds and both UP-GMM($\ell_1$) and SPLITUP($\ell_1$) are consistent, we introduce two controlled violations. Let $E \in [m]$ denote the environment label, so that the one-hot instrument is $I = e_E$, and let $\mu_j \in \mathbb{R}^d$ denote the baseline mean of $X$ in environment $j$.

First, we violate the exclusion restriction (Assumption 1 (ii)) by adding an environment-specific direct effect to the outcome,

$$Y \leftarrow Y + \alpha_{\text{excl}}v_E, \qquad v_j \overset{\text{i.i.d.}}{\sim} \mathcal{N}(0,1), \qquad j \in [m],$$

where $\alpha_{\text{excl}} \geqslant 0$ controls the strength of the violation. This induces $\text{Cov}(I, \varepsilon) \neq 0$. Second, we violate the cross-sample moment invariance (Assumption 1 (i)) by perturbing the environment means on the $\tilde{X}$-side,

$$\mu_j^{\tilde{X}} = \mu_j + \alpha_{\text{cov}}\Delta_j, \qquad \Delta_j \in \mathbb{R}^d, \qquad \Delta_{jk} \overset{\text{i.i.d.}}{\sim} \mathcal{N}(0, \sigma_{\text{env}}^2),$$

where $\alpha_{\text{cov}} \geqslant 0$ controls the strength of the covariance mismatch and $\sigma_{\text{env}}$ is the standard deviation used for the baseline environment means. This induces $\text{Cov}(I, X) \neq \text{Cov}(\tilde{I}, \tilde{X})$. Setting the corresponding violation strength to zero recovers the well-specified baseline.

Figure 7 shows that estimation error increases smoothly with the violation strength.

### D. Categorical Instruments: Environments without Features as Instruments

Assume $I$ is categorical with $m \in \mathbb{N}$ domains and is encoded one–hot as $I \in \{e_1, \dots, e_m\} \subset \mathbb{R}^m$, where $e_k$ is the $k$th standard basis vector. Suppose that the domain frequencies match across the two samples, with all domains equally likely: $\mathbb{P}(I = e_k) = \mathbb{P}(\tilde{I} = e_k) = \frac{1}{m}$ for all $k \in [m]$. Let $n_k = \sum_{i=1}^{n} \mathbb{1}\{I_i = e_k\}$ and $\tilde{n}_k = \sum_{j=1}^{\tilde{n}} \mathbb{1}\{\tilde{I}_j = e_k\}$ denote the domain counts in the $Y-$ and $\tilde{X}-$samples, respectively, and define the within–domain sample means

$$\bar{Y}_k = \frac{1}{n_k} \sum_{i:I_i=e_k} Y_i, \qquad \bar{X}_k = \frac{1}{\tilde{n}_k} \sum_{j:\tilde{I}_j=e_k} \tilde{X}_j.$$

under equal domain probabilities and matched empirical shares, $n_k/n = \tilde{n}_k/\tilde{n} = 1/m$ for all $k$. Then TS-2SLS, TS-IV and UP-GMM ($\hat{\beta}_{\text{GMM}}^{\text{UP}}(\text{Id})$) are the same estimator and are equivalent to OLS on the category means:

$$\hat{\beta}_{\text{GMM}}^{\text{UP}}(\text{Id}) = \Big( \sum_{k=1}^{m} \bar{X}_k \bar{X}_k^{\top} \Big)^{-1} \Big( \sum_{k=1}^{m} \bar{X}_k \bar{Y}_k \Big).$$

# E. Inference on the Estimated Support

The Lasso is used only for support recovery. Under beta–min and regularity conditions, $\widehat{S}$ is consistent: $\mathbb{P}(\widehat{S} = S^*) \to 1$. We then refit an unpenalized GMM on $\widehat{S}$ to obtain $\tilde{\beta}$, reducing to a standard finite-dimensional GMM problem with asymptotic normality and sandwich standard errors (Hansen, 1982). Since mis-selection becomes negligible under a beta-min condition, usual Wald intervals based on the refitted GMM standard errors are asymptotically valid. We make this precise in Theorem E.1.

**Theorem E.1** (Oracle CIs on the estimated support). *Assume Assumption 3 and the beta-min condition from Theorem 4.2 for $W_N$ and let $\hat{S}(W_N)$ be defined as in Theorem 4.2. Assume that there is a second sequence of weight matrices $W'_N \overset{p}{\to} W'_0 \succ 0$. Define*

$$\tilde{\beta}(W'_N) := \underset{\beta \in \mathbb{R}^d : \beta_{\widehat{S}^c(W_N)} = 0}{\arg\min} (\widehat{\mathrm{Cov}}(I, Y) - \widehat{\mathrm{Cov}}(\tilde{I}, \tilde{X})\beta)^\top$$
$$\cdot W'_N (\widehat{\mathrm{Cov}}(I, Y) - \widehat{\mathrm{Cov}}(\tilde{I}, \tilde{X})\beta),$$

*and*

$$V_{S^*} := (\mathrm{Cov}(I, X)_{S^*}^\top W'_0 \mathrm{Cov}(I, X)_{S^*})^{-1}$$
$$\cdot \left( \mathrm{Cov}(I, X)_{S^*}^\top W'_0 \Omega W'_0 \mathrm{Cov}(I, X)_{S^*} \right)$$
$$\cdot (\mathrm{Cov}(I, X)_{S^*}^\top W'_0 \mathrm{Cov}(I, X)_{S^*})^{-1},$$

*where $\Omega := \mathrm{Var}(\sqrt{N}\hat{g}_N(\beta^*))$. We have that*

$$\sqrt{N}(\tilde{\beta}(W'_N) - \beta^*) \overset{d}{\to} \mathcal{N}(0, \tilde{V}),$$

*where $\tilde{V}$ has the $S^* \times S^*$ block equal to $V_{S^*}$ and is zero elsewhere.*

*Remark* E.2. The asymptotic variance $V_{S^*}$ only depends on $W'_N$ and not on $W_N$. We are therefore free to choose $W_N = \mathrm{Id}_m$. We define $W'_N := \hat{\Omega}_{\hat{S}(\mathrm{Id}_m)}^{-1} \in \mathbb{R}^{m \times m}$, where $\hat{\Omega}_{\hat{S}(\mathrm{Id}_m)}$ is defined as in (K.1) but we consider all variables restricted to $\hat{S}(\mathrm{Id}_m)$. If $\hat{\Omega}_{\hat{S}(\mathrm{Id}_m)}$ is not positive definite, we choose $W'_N := \mathrm{Id}_m$ (we do this to ensure that the estimator is well-defined). Since $\hat{S}(\mathrm{Id}_m) \overset{p}{\to} S^*$ we have $W'_N \overset{p}{\to} \Omega_{S^*}^{-1}$ and therefore this choice minimizes the asymptotic variance.

# F. Variance Reduction in Practice

In finite samples, cross-fitting stabilizes the denominator by removing the leading measurement-error bias, but it can introduce additional variability through random splitting. We therefore use two simple variance-reduction devices that preserve consistency and leave the preceding theory unchanged. All methods described in this section are detailed in Appendix G.

**Monte Carlo averaging over random splits.** To reduce the finite-sample variance introduced by sample splitting, we repeat the random split multiple times and average the resulting estimates of $C_{XX}$. Each split-specific estimator is consistent for the same population target, hence their average remains consistent. Consequently, all results from Section 4 apply verbatim to the averaged estimator.

**Closed form for the infinite-split average.** A natural question is whether the split-average admits a closed form when we average over *all* possible splits. The answer is yes. Fix the $(\tilde{I}, \tilde{X})$ sample and consider $K = 2$ splits into two equal halves. The cross-fit denominator for a split $(A, B)$ is

$$C_{XX}(A, B) := m \, \widehat{\mathrm{Cov}}_A(\tilde{I}, \tilde{X})^\top \widehat{\mathrm{Cov}}_B(\tilde{I}, \tilde{X}).$$

If we draw $H \in \mathbb{N}$ i.i.d. random splits $(A_1, B_1), \ldots, (A_H, B_H)$ and let $H \to \infty$, the Monte Carlo average converges (conditional on the data) to the conditional expectation over a uniform random split. This expectation has the closed form

$$
\begin{aligned}
\frac{1}{m} \bar{C}_{XX} &:= \lim_{H \to \infty} \frac{1}{H} \sum_{h=1}^{H} C_{XX}(A_h, B_h) \\
&= \frac{1}{\tilde{n}(\tilde{n}-1)} \sum_{i \neq j} (\tilde{I}_i \tilde{X}_i^\top)^\top (\tilde{I}_j \tilde{X}_j^\top) \\
&= \frac{\tilde{n}}{\tilde{n}-1} \widehat{\mathrm{Cov}}(\tilde{I}, \tilde{X})^\top \widehat{\mathrm{Cov}}(\tilde{I}, \tilde{X}) \\
&\quad - \frac{1}{\tilde{n}(\tilde{n}-1)} \sum_{i=1}^{\tilde{n}} (\tilde{I}_i \tilde{X}_i^\top)^\top (\tilde{I}_i \tilde{X}_i^\top).
\end{aligned}
$$

Thus, the $H \to \infty$ split-average equals the usual plug-in quadratic form $\widehat{\mathrm{Cov}}(\tilde{I}, \tilde{X})^\top \widehat{\mathrm{Cov}}(\tilde{I}, \tilde{X})$ minus a diagonal 'self-inner-product' correction. This correction is what removes the measurement-error bias.

*Proof.* Write $\widehat{\mathrm{Cov}}_A(\tilde{I}, \tilde{X}) = \frac{2}{n} \sum_{i \in A} g_i$ and $\widehat{\mathrm{Cov}}_B(\tilde{I}, \tilde{X}) = \frac{2}{n} \sum_{j \in B} g_j$, hence

$$
\widehat{\mathrm{Cov}}_A(\tilde{I}, \tilde{X})^\top \widehat{\mathrm{Cov}}_B(\tilde{I}, \tilde{X}) = \frac{4}{n^2} \sum_{i \in A} \sum_{j \in B} g_i^\top g_j.
$$

For a uniform random split, each ordered pair $(i, j)$ with $i \neq j$ lands in $(A, B)$ with probability

$$
\mathbb{P}(i \in A, j \in B) = \frac{n/2}{n} \cdot \frac{n/2}{n-1} = \frac{n}{4(n-1)}.
$$

Taking conditional expectation over the split therefore gives

$$
\mathbb{E}\big[ \widehat{\mathrm{Cov}}_A(\tilde{I}, \tilde{X})^\top \widehat{\mathrm{Cov}}_B(\tilde{I}, \tilde{X}) \mid (g_i)_{i=1}^n \big] = \frac{4}{n^2} \cdot \frac{n}{4(n-1)} \sum_{i \neq j} g_i^\top g_j = \frac{1}{n(n-1)} \sum_{i \neq j} g_i^\top g_j,
$$

and rewriting $\sum_{i \neq j} g_i^\top g_j = (\sum_i g_i)^\top (\sum_j g_j) - \sum_i g_i^\top g_i$ yields the displayed closed form. $\qquad \square$

**Categorical improvements via stratification.** When the instrument is categorical (e.g., a one-hot encoding of environments), we can reduce the finite-sample variance of the cross-fit denominator by *stratifying the folds by environment*. Concretely, instead of splitting the pooled $(\tilde{I}, \tilde{X})$ sample uniformly at random, we split *within each environment* and then form folds by concatenating the corresponding within-environment parts. This ensures that every fold has (approximately) the same environment composition, avoiding the additional variability that arises when unstratified folds differ in their environment proportions or omit small environments entirely. We use this stratification both for the split estimator (and optionally average over several stratified redraws) and for the analytic estimator by taking the $H \to \infty$ limit corresponding to stratified splits, which can be computed efficiently from per-environment sufficient statistics. Stratification does not change the population target of $C_{XX}$; it only stabilizes the finite-sample cross-moments by controlling fold composition. Details are given in Appendix G, Algorithm 6 and Algorithm 4.

**Closed form for stratified categorical moments.** Assume $\tilde{I}$ is one-hot with environment label $e(i) \in [m]$. Let $J_e := \{i : e(i) = e\}$, $r_e := |J_e|$, and restrict to environments with $r_e \geqslant 2$ (denote the kept set by $\mathcal{E}$). For each $e \in \mathcal{E}$, define

$$
S_e := \sum_{i \in J_e} \tilde{X}_i, \qquad Q_e := \sum_{i \in J_e} \tilde{X}_i \tilde{X}_i^\top, \qquad M_e := \frac{S_e S_e^\top - Q_e}{r_e(r_e - 1)} = \frac{1}{r_e(r_e - 1)} \sum_{\substack{i, j \in J_e \\ i \neq j}} \tilde{X}_i \tilde{X}_j^\top.
$$

Consider $K = 2$ *stratified* splits where, independently for each environment $e$, we split $J_e$ into two disjoint parts $A_e, B_e$ of sizes $a_e := |A_e|$ and $b_e := |B_e|$ with $a_e + b_e = r_e$ (e.g., $a_e = \lfloor r_e/2 \rfloor$, $b_e = \lceil r_e/2 \rceil$), and let $A := \bigcup_e A_e$, $B := \bigcup_e B_e$ with $n_A := |A| = \sum_e a_e$ and $n_B := |B| = \sum_e b_e$. Using one-hotness, only within-environment cross-pairs contribute

---

**Algorithm 1** TS-IV (ridge-stabilized)

---

1: **Input:** $\{(I_i, Y_i)\}_{i=1}^n$, $\{(\tilde{I}_j, \tilde{X}_j)\}_{j=1}^{\tilde{n}}$, ridge $\lambda > 0$
2: Compute $a := \widehat{\mathrm{Cov}}(I, Y) \in \mathbb{R}^m$
3: Compute $B := \widehat{\mathrm{Cov}}(\tilde{I}, \tilde{X}) \in \mathbb{R}^{m \times d}$
4: Solve $(B^\top B + \lambda \mathrm{Id}_d)\hat{\beta} = B^\top a$
5: **Output:** $\hat{\beta}$

---

---

**Algorithm 2** TS-2SLS (ridge-stabilized)

---

1: **Input:** $\{(I_i, Y_i)\}_{i=1}^n$, $\{(\tilde{I}_j, \tilde{X}_j)\}_{j=1}^{\tilde{n}}$, ridge $\lambda > 0$
2: First stage: $\Gamma := (\tilde{I}^\top \tilde{I} + \lambda I_m)^{-1} \tilde{I}^\top \tilde{X}$
3: Predict in $Y$-sample: $\hat{X} := I\Gamma$
4: Second stage: solve $(\hat{X}^\top \hat{X} + \lambda \mathrm{Id}_d)\hat{\beta} = \hat{X}^\top Y$
5: **Output:** $\hat{\beta}$

---

to $\widehat{\mathrm{Cov}}_A(\tilde{I}, \tilde{X})^\top \widehat{\mathrm{Cov}}_B(\tilde{I}, \tilde{X})$. Moreover, for any ordered pair $i \neq j$ in the same environment $e$, the probability that $(i, j)$ lands in $(A, B)$ under a stratified split equals

$$\mathbb{P}(i \in A, \, j \in B \mid e(i) = e(j) = e) = \frac{a_e}{r_e} \cdot \frac{b_e}{r_e - 1}.$$

Taking conditional expectation over the stratified split and collecting terms yields the analytic $H \to \infty$ limit

$$\frac{1}{m}\bar{C}_{XX}^{\mathrm{strat}} := \mathbb{E}\Big[\widehat{\mathrm{Cov}}_A(\tilde{I}, \tilde{X})^\top \widehat{\mathrm{Cov}}_B(\tilde{I}, \tilde{X}) \,\Big|\, (\tilde{I}_i, \tilde{X}_i)_{i=1}^{\tilde{n}}\Big] = \frac{1}{n_A n_B} \sum_{e \in \mathcal{E}} \frac{a_e b_e}{r_e(r_e - 1)}\big(S_e S_e^\top - Q_e\big).$$

Equivalently, $\frac{1}{m}\bar{C}_{XX}^{\mathrm{strat}} = \frac{1}{n_A n_B} \sum_{e \in \mathcal{E}} a_e b_e \, M_e$. This is the closed form used in Algorithm 6.

## G. Algorithm Details

In total we consider 6 algorithms: TS-IV, TS-2SLS, UP-GMM, SPLITUP, SPLITUP (analytic) and naive OLS. For each method, we give a short intuitive description and a compact pseudocode block.

**TS-IV.** TS-IV solves the unpaired moment equation $\mathrm{Cov}(I, Y) = \mathrm{Cov}(\tilde{I}, \tilde{X})\beta$ by plugging in empirical cross-covariances from the two samples. For numerical stability we use a small ($10^{-10}$) ridge-stabilization. Note that UP-GMM with $W_N = \mathrm{Id}$ is equivalent to TS-IV. The pseudocode for TS-IV is given in Algorithm 1.

**TS-2SLS.** TS-2SLS is two-stage least squares with unpaired data: first learn the mapping from instruments to covariates on the $(\tilde{I}, \tilde{X})$ sample, then use instruments in the $(I, Y)$ sample to predict the missing covariates, and finally regress $Y$ on the predicted covariates. We again add a small ridge penalty ($10^{-10}$) for numerical stability. The pseudocode for TS-2SLS is given in Algorithm 2.

**UP-GMM.** UP-GMM is a GMM estimator for the same unpaired moment condition as TS-IV, but it optionally uses an estimated optimal weight matrix and includes $\ell_1$-regularization for settings with sparse $\beta^*$. Intuitively, some moment coordinates are noisier than others because they come from two different samples; optimal weighting downweights noisy moments. A sparse variant adds an $\ell_1$ penalty and can refit on the selected support. We add a small ridge penalty ($10^{-10}$) for numerical stability. For all experiments we set set `optimal weight = True` and if $\beta^*$ is sparse we additionally set `l1 = True` and `post-refit = True`. The pseudocode for UP-GMM is given in Algorithm 3.

**SPLITUP.** In the high-dimensional instrument regime, the plug-in denominator $B^\top B$ has measurement-error bias that does not vanish with $m$. SPLITUP removes this by forming a cross-moment denominator from independent folds of the $(\tilde{I}, \tilde{X})$ sample. It then solves $C_{XX}\beta = C_{XY}$, optionally with $\ell_1$ regularization and post-refit. We add a small ridge penalty ($10^{-10}$) for numerical stability. For all experiments we choose $K = 2$, $H = 10$, and for sparse $\beta^*$ we additionally set `l1 = True` and `post-refit = True`. The pseudocode for SPLITUP is given in Algorithm 5.

---

**Algorithm 3** UP-GMM

---

1: **Input:** ridge penalty $\lambda 2 > 0$, lasso penalty $\lambda 1 > 0$, option `optimal_weight`, option `l1`, option `post-refit`
2: $a := \widehat{\mathrm{Cov}}(I, Y)$, $B := \widehat{\mathrm{Cov}}(\tilde{I}, \tilde{X})$
3: Init: $\hat{\beta}^{(0)} := (B^\top B + \lambda \mathrm{Id}_d)^{-1} B^\top a$
4: **if** `optimal_weight` **then**
5:     Estimate moment covariance $\hat{\Omega}$ using $IY$ and $\tilde{I}(\tilde{X}^\top \hat{\beta}^{(0)})$                *(see (K.1))*
6:     Set $W := (\hat{\Omega} + \lambda_2 I_m)^{-1}$
7: **else**
8:     Set $W := I_m$
9: **end if**
10: **if** `l1` is false **then**
11:     Solve $(B^\top W B + \lambda_2 \mathrm{Id}_d)\hat{\beta} = B^\top W a$
12: **else**
13:     Solve $\hat{\beta} \in \arg\min_\beta \frac{1}{2}\|W^{1/2}(a - B\beta)\|_2^2 + \lambda_1 \|\beta\|_1$
14:     **if** `post-refit` **then**
15:         Let $\hat{S} := \{j : |\hat{\beta}_j| > 0\}$ and refit dense UP-GMM restricted to $\hat{S}$
16:     **end if**
17: **end if**
18: **Output:** $\hat{\beta}$

---

**SPLITUP (analytic).** SPLITUP (analytic) replaces Monte Carlo splitting by the closed-form infinite-split limit of the cross-fit denominator. Intuitively, it equals the usual quadratic form minus a self-inner-product correction that removes the leading measurement-error bias. We add a small ridge penalty ($10^{-10}$) for numerical stability. For sparse $\beta^*$ we set `l1 = True` and `post-refit = True`. The pseudocode for SPLITUP is given in Algorithm 7.

**Naive OLS.** Naive OLS randomly pairs $\tilde{X}$ rows with $Y$ rows and runs OLS. This ignores the missing joint structure and does not target the IV moment condition; it serves only as a baseline.

## H. Data Generating Processes

All synthetic experiments follow the unpaired IV model in (2.1). We generate two independent samples: a $Y$-sample $\{(I_i, Y_i)\}_{i=1}^n$ and an $X$-sample $\{(\tilde{I}_j, \tilde{X}_j)\}_{j=1}^{\tilde{n}}$, where $X$ is latent in the $Y$-sample and $\tilde{Y}$ is latent in the $X$-sample. Hidden confounding is introduced via a latent variable $U$ that affects both $X$ and $Y$. Throughout, we use balanced sample sizes of the form

$$n = mr, \qquad \tilde{n} = m\tilde{r},$$

where $m$ denotes the number (or dimension) of instruments and $r, \tilde{r}$ control the sample-to-instrument ratios. In all settings, the two samples share the same first-stage cross-moment (Assumption 1.(i)) by construction.

**Common structural equations.** We generate latent covariates and outcomes via

$$X = \mu(I) + \gamma_x U + \varepsilon_x, \qquad Y = X^\top \beta^* + \gamma_y U + \varepsilon_y, \qquad \tilde{X} = \mu(\tilde{I}) + \gamma_x \tilde{U} + \tilde{\varepsilon}_x,$$

where $U, \tilde{U} \sim \mathcal{N}(0, \sigma_u^2)$ is an unobserved confounder and $\tilde{\varepsilon}_x, \varepsilon_x, \varepsilon_y$ are mean-zero noise terms. The dependence of $Y$ on $U$ implies endogeneity ($\mathbb{E}[\varepsilon \mid X] \neq 0$), while exogeneity with respect to the instrument is enforced by construction ($\mathbb{E}[\varepsilon \mid I] = 0$).

### H.1. Categorical Instruments (one-hot Environments)

**Instrument.** For categorical instruments, $I \in \{e_1, \ldots, e_m\} \subset \mathbb{R}^m$ is one-hot with uniform environment probability. The $Y$-sample and $X$-sample are balanced across environments: for each environment $e \in [m]$ we generate exactly $r$ observations in the $Y$-sample and $\tilde{r}$ observations in the $X$-sample.

---

**Algorithm 4** FOLDCROSSCOV: compute $B_k = \widehat{\mathrm{Cov}}_k(\tilde{I}, \tilde{X})$

---

1: **Input:** fold data $\{(\tilde{I}_j, \tilde{X}_j)\}_{j \in F_k}$ with $|F_k| = n_k$
2: **Output:** $B_k \in \mathbb{R}^{m \times d}$
3: **if** $\tilde{I}$ is one-hot (categorical) **then**
4:     Compute fold mean $\bar{X} := \frac{1}{n_k} \sum_{j \in F_k} \tilde{X}_j$
5:     **for** each environment $e \in [m]$ **do**
6:         $J_{k,e} := \{j \in F_k : \tilde{I}_j = e_e\}, \quad n_{k,e} := |J_{k,e}|, \quad p_{k,e} := n_{k,e}/n_k$
7:         **if** $n_{k,e} > 0$ **then**
8:             $\bar{X}_{k,e} := \frac{1}{n_{k,e}} \sum_{j \in J_{k,e}} \tilde{X}_j$
9:         **else**
10:            $\bar{X}_{k,e} := \bar{X}$
11:         **end if**
12:         Set row $e$ of $B_k$ as $(B_k)_{e,:} := p_{k,e} (\bar{X}_{k,e} - \bar{X})^\top$
13:     **end for**
14: **else**
15:     Center columns: $\tilde{I}_c := \tilde{I} - \frac{1}{n_k} \sum_{j \in F_k} \tilde{I}_j, \quad \tilde{X}_c := \tilde{X} - \frac{1}{n_k} \sum_{j \in F_k} \tilde{X}_j$
16:     $B_k := \frac{1}{n_k} \tilde{I}_c^\top \tilde{X}_c$
17: **end if**
18: **return** $B_k$

---

**Environment-specific first stage and heteroskedasticity.** We draw environment means $\mu_e \in \mathbb{R}^d$ i.i.d. as

$$\mu_e \sim \mathcal{N}(0, \mathrm{Id}_d), \qquad e \in [m].$$

To introduce realistic heteroskedasticity, we draw environment-specific noise scales

$$\sigma_{x,e} \sim \sigma_x \cdot \mathrm{LogNormal}(0, 0.5), \qquad \sigma_{\varepsilon,e} \sim \sigma_\varepsilon \cdot \mathrm{LogNormal}(0, 0.5),$$

clipped to a fixed range and renormalized to keep the average scale constant.

**Sampling.** For each $Y$-sample observation in environment $e$ we sample

$$U \sim \mathcal{N}(0, \sigma_u^2), \quad \varepsilon_x \sim \mathcal{N}(0, \sigma_{x,e}^2 \mathrm{Id}_d), \quad \varepsilon_y \sim \mathcal{N}(0, \sigma_{\varepsilon,e}^2),$$

set $X_{\mathrm{lat}} = \mu_e + \gamma_x U + \varepsilon_x$, and output

$$Y = X_{\mathrm{lat}}^\top \beta^* + \gamma_y U + \varepsilon_y.$$

For each $X$-sample observation in the same environment $e$ we sample $\tilde{U}$ and $\tilde{\varepsilon}_x$ analogously and output

$$\tilde{X} = \mu_e + \gamma_x \tilde{U} + \tilde{\varepsilon}_x.$$

This construction ensures $\mathrm{Cov}(I, X) = \mathrm{Cov}(\tilde{I}, \tilde{X})$.

**Setting 1 (categorical): finite-dimensional instruments, sparse $\beta^*$.** We fix the number of environments $m$ and dimension $d$, and choose $\beta^*$ to be sparse with $s^*$ nonzeros:

$$\|\beta^*\|_0 = s^*, \qquad (\beta^*)_j \in [-1, -0.5] \cup [0.5, 1] \text{ uniformly.}$$

We increase the sample size through $r, \tilde{r}$ (equivalently $n, \tilde{n} \to \infty$ with fixed $m$), matching the finite-dimensional instrument regime.

**Setting 2 (categorical): high-dimensional instruments, dense $\beta^*$.** We consider the high-dimensional instrument regime by increasing $m$ while keeping the ratios $r = n/m$ and $\tilde{r} = \tilde{n}/m$ fixed. We consider $d$. The causal effect $\beta^*$ is dense ($(\beta^*)_j \in [-1, -0.5] \cup [0.5, 1]$ uniformly), and performance is reported as a function of $n/m$. This is the regime where the plug-in denominator in unpaired IV exhibits persistent measurement-error bias. This data generating process has also been used to generate the data underlying Figure 1 in Section 1.

---

---

**Algorithm 5** SPLITUP

---

1: **Input:** data, folds $K \geqslant 2$, redraws $H$, ridge $\lambda_2 > 0$, lasso $\lambda_1 > 0$, option `l1`, option `post-refit`, optimal weight
2: $a := \widehat{\mathrm{Cov}}(I, Y)$, $B := \widehat{\mathrm{Cov}}(\tilde{I}, \tilde{X})$
3: $C_{XY} := m\, B^\top a$
4: Initialize $C_{XX} := 0$
5: **for** $b = 1$ to $H$ **do**
6:     Split $\{1, \ldots, \tilde{n}\}$ into $K$ folds (stratify within environments if $I$ is one-hot)
7:     For each fold $k$, compute $B_k := \text{FOLDCROSSCOV}\left(\{(\tilde{I}_j, \tilde{X}_j)\}_{j \in F_k}\right)$ (Algorithm 4)
8:     $C_{XX} := C_{XX} + \frac{m}{K(K-1)} \sum_{h \neq k} B_h^\top B_k$
9: **end for**
10: $C_{XX} := \frac{1}{H} C_{XX}$
11: **if** optimal weight **then**
12:     Estimate moment covariance $\hat{\Omega}$
13:     Set $W := (\hat{\Omega} + \lambda_2 \mathrm{Id}_d)^{-1}$
14: **else**
15:     Set $W := \mathrm{Id}_d$
16: **end if**
17: **if** `l1` is false **then**
18:     Solve $(C_{XX}^\top W C_{XX} + \lambda_2 \mathrm{Id}_d)\hat{\beta} = C_{XX}^\top W C_{XY}$
19: **else**
20:     Solve $\hat{\beta} \in \arg\min_\beta \frac{1}{2} \|W^{1/2}(C_{XY} - C_{XX}\beta)\|_2^2 + \lambda_1 \|\beta\|_1$
21:     **if** `post-refit` **then**
22:         Let $\hat{S} := \{j : |\hat{\beta}_j| > 0\}$ and refit dense SPLITUP on the subset $\hat{S}$
23:     **end if**
24: **end if**
25: **Output:** $\hat{\beta}$

---

**Setting 3 (categorical): high-dimensional instruments, sparse $\beta^*$.** This setting combines high-dimensional instruments ($m \to \infty$ with fixed $n/m$) and sparse causal effects ($\|\beta^*\|_0 = s^*$). To induce weak and low-rank first stages, we replace the i.i.d. environment means by a low-rank construction:

$$Z_e \sim \mathcal{N}(0, \mathrm{Id}_k), \qquad \mu_e = Z_e A^\top, \qquad A \in \mathbb{R}^{d \times k} \text{ fixed}, \ k \ll d.$$

Equivalently, $\{\mu_e\}_{e=1}^m$ lie in the $k$-dimensional subspace spanned by the columns of $A$, yielding a rank-constrained moment matrix and making sparsity essential for identification. Noise scales and confounding are generated as in Setting 1.

## H.2. Continuous Instruments

**Instrument and first-stage map.** For continuous instruments, we draw (independently)

$$I \in \mathbb{R}^m, \qquad I \sim \mathcal{N}\left(0, \frac{1}{m}\mathrm{Id}_m\right), \qquad \tilde{I} \in \mathbb{R}^m, \qquad \tilde{I} \sim \mathcal{N}\left(0, \frac{1}{m}\mathrm{Id}_m\right),$$

and generate covariates through a linear first stage with a shared matrix $\Pi \in \mathbb{R}^{m \times d}$:

$$X = I\Pi + \gamma_x U + \varepsilon_x, \qquad Y = X^\top \beta^* + \gamma_y U + \varepsilon_y, \qquad \tilde{X} = \tilde{I}\Pi + \gamma_x \tilde{U} + \tilde{\varepsilon}_x.$$

The matrix $\Pi$ is sampled once per dataset and shared across the $Y$-sample and $\tilde{X}$-sample, ensuring $\mathrm{Cov}(I, X) = \mathrm{Cov}(\tilde{I}, \tilde{X})$ by construction.

**Setting 1 (continuous): finite-dimensional instruments, sparse $\beta^*$.** We fix $m$ and $d$, choose $\beta^*$ sparse with $\|\beta^*\|_0 = s^*$, and increase sample size via $r, \tilde{r}$ (thus $n, \tilde{n} \to \infty$ with fixed $m$). The first-stage matrix $\Pi$ is dense i.i.d. Gaussian, which yields a full-rank population first stage when $m$ is sufficiently large, but the regime of interest keeps $d$ larger than $m$ so sparse structure is required.

---

**Algorithm 6** CROSSFOLDDENOM: compute $C_{XX}$ via analytic cross-pair moments

---

1: **Input:** $\{(\tilde{I}_j, \tilde{X}_j)\}_{j=1}^{\tilde{n}}$ with $\tilde{I}_j \in \mathbb{R}^m$, $\tilde{X}_j \in \mathbb{R}^d$
2: **Output:** $C_{XX} \in \mathbb{R}^{d \times d}$
3: **if** $\tilde{I}$ is one-hot (categorical) **then**
4:      For each environment $e \in [m]$, let $J_e := \{j : \tilde{I}_j = e_e\}$ and $r_e := |J_e|$
5:      Initialize $n_A := 0$, $n_B := 0$, and $A := 0_{d \times d}$
6:      **for** each $e \in \mathcal{E}$ **do**
7:          Choose split sizes $a_e := \lfloor r_e/2 \rfloor$ and $b_e := r_e - a_e$
8:          $S_e := \sum_{j \in J_e} X_j$
9:          $Q_e := \sum_{j \in J_e} X_j X_j^\top$
10:         $A := A + \frac{a_e b_e}{r_e(r_e-1)}(S_e S_e^\top - Q_e)$
11:         $n_A := n_A + a_e, \quad n_B := n_B + b_e$
12:      **end for**
13:      $C_{XX} := m \cdot \frac{1}{n_A n_B} A$
14: **else**
15:      **generic branch:**
16:      Form matrices $I \in \mathbb{R}^{\tilde{n} \times m}$ and $X \in \mathbb{R}^{\tilde{n} \times d}$ whose rows are $I_j^\top$ and $X_j^\top$
17:      $S := I^\top X$
18:      term1 $:= S^\top S$
19:      $w_j := \|I_j\|_2^2$ for $j \in [\tilde{n}]$
20:      term2 $:= \sum_{j=1}^{\tilde{n}} w_j X_j X_j^\top$
21:      scale $:= 1/\max(\tilde{n}(\tilde{n}-1), 1)$
22:      $C_{XX} := m\,\text{scale}\,(\text{term1} - \text{term2})$
23: **end if**
24: **return** $C_{XX}$

---

**Setting 2 (continuous): high-dimensional instruments, dense $\beta^*$.** We increase $m$ while keeping $n/m$ and $\tilde{n}/m$ fixed and use a dense $\beta^*$. The persistence of measurement-error bias for plug-in denominators in two-sample IV carries over to this continuous-instrument regime, motivating cross-moment denominators.

**Setting 3 (continuous): high-dimensional instruments, sparse $\beta^*$.** We again take $m \to \infty$ with fixed ratios $n/m$ and $\tilde{n}/m$, but use a sparse $\beta^*$ with $\|\beta^*\|_0 = s^*$. To create a low-rank first stage (analogous to the low-rank environment means in the categorical case), we draw a fixed matrix $A \in \mathbb{R}^{d \times k}$ with $k \ll d$ and set

$$Z \in \mathbb{R}^{m \times k}, \quad Z_\ell \sim \mathcal{N}(0, \text{Id}_k), \qquad \Pi := \pi_{\text{scale}}(ZA^\top),$$

so $\Pi$ has rank at most $k$. Confounding is generated as in setting 2.

### H.3. Choices of Constants

For all experiments, we set $\tilde{n} = n$ and $\tilde{r} = r$ and

$$\gamma_x := \frac{1}{5} \qquad \gamma_y := \frac{1}{5} \qquad \sigma_u := \frac{1}{5} \qquad \sigma_x := 1 \qquad \sigma_\varepsilon := \frac{1}{5}$$

We use the same parameters for the categorical instrument and the continuous instrument experiments.

**Setting 1.** We set $m = 100$, $d = 200$ and $s^* = 10$.

**Setting 2.** We set $d = 2$.

**Setting 3.** We set $k = 60$, $d = 100$ and $s^* = 10$.

For sparse UP-GMM, we set $\lambda_1 = \alpha_{\max}\sqrt{\log(d+1)/m}$, where $\alpha_{\max}$ is the smallest lasso penalty that sets all coefficients to zero for the weighted and column-normalized GMM design. For sparse SPLITUP, we use $\lambda_1 = \sqrt{\log(d+1)/m}$ after

---

**Algorithm 7** SPLITUP (analytic)

---

1: **Input:** data, ridge $\lambda_2 > 0$, lasso $\lambda_1 > 0$, option `l1`, option `post-refit`, `optimal weight`
2: Compute $C_{XY} := m\,\widehat{\mathrm{Cov}}(\tilde{I}, \tilde{X})^\top \widehat{\mathrm{Cov}}(I, Y)$
3: $C_{XX} := \mathrm{CROSSFOLDDENOM}\big(\{(\tilde{I}_j, \tilde{X}_j)\}_{j=1}^{\tilde{n}}\big)$ (Algorithm 6)
4: **if** `optimal weight` **then**
5:     Estimate moment covariance $\hat{\Omega}$
6:     Set $W := (\hat{\Omega} + \lambda_2 I_d)^{-1}$
7: **else**
8:     Set $W := I_d$
9: **end if**
10: **if** `l1` is false **then**
11:     Solve $(C_{XX}^\top W C_{XX} + \lambda_2 \mathrm{Id}_d)\hat{\beta} = C_{XX}^\top W C_{XY}$
12: **else**
13:     Solve $\hat{\beta} \in \arg\min_\beta \frac{1}{2}\|W^{1/2}(C_{XY} - C_{XX}\beta)\|_2^2 + \lambda_1\|\beta\|_1$
14:     **if** `post-refit` **then**
15:         Let $\hat{S} := \{j : |\hat{\beta}_j| > 0\}$ and refit dense SPLITUP (analytic) on the subset $\hat{S}$
16:     **end if**
17: **end if**
18: **Output:** $\hat{\beta}$

---

---

**Algorithm 8** Naive OLS via random pairing

---

1: **Input:** $\{Y_i\}_{i=1}^n$, $\{\tilde{X}_j\}_{j=1}^{\tilde{n}}$
2: Set $n' := \min(n, \tilde{n})$ and sample indices $\mathcal{I}_X, \mathcal{I}_Y$ of size $n'$
3: Form paired matrices $X_p := \tilde{X}_{\mathcal{I}_X}$ and $Y_p := Y_{\mathcal{I}_Y}$
4: Center columns of $X_p$ and center $Y_p$
5: Output $\hat{\beta} := (X_p^\top X_p)^\dagger X_p^\top Y_p$

---

diagonal normalization of $C_{XX}$. In both cases, we refit without the penalty on the selected support. For a data-driven procedure to select the $\ell_1$ penalty, see Appendix I.5.

### H.4. Causal Chambers Light Tunnel

We evaluate our algorithms on the causal chambers proposed by Gamella et al. (2025), a computer-controlled physical system that enables reproducible, automated data collection with control over actuators and sensors. We use the light-tunnel chamber in the standard configuration (no camera), which contains a controllable light source and multiple photodiode sensors positioned along the tunnel. Our experiment uses a subset of actuators and infrared (IR) sensor readings. We explain the variables and mapping to the variable names in Gamella et al. (2025) in Table 2. Figure 9 is a schematic of the chamber including variable names.

| Causal role | Symbol | Variable (Gamella et al., 2025) | Physical function |
|---|---|---|---|
| Instrument | $I$ | `led_1_ir` | IR LED drive level for LED 1 |
| Confounder | $U$ | `red` | intensity of red channel (source light) |
| Treatment | $X$ | `ir_1` | IR photodiode reader for LED 1 |
| Mediator | $M$ | `led_2_uv` | UV LED drive level for LED 2 |
| Outcome | $Y$ | `ir_2` | IR photodiode reader for LED 2 |

*Table 2.* Variable names, their description and the corresponding names in Gamella et al. (2025). Note that Gamella et al. (2025) only talk about two LEDs for each sensor (`l_11` and `l_12`). We use the newer version of the causal chambers where the LEDs are renamed: `led_1_uv` and `led_1_ir` for sensor 1 and `led_2_uv` and `led_2_ir` for sensor 2. The overall structure of the newer version of the causal chamber is the same as the one described by Gamella et al. (2025).

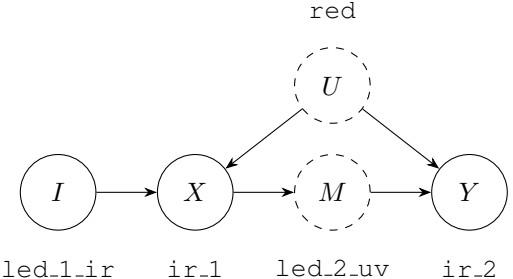

*Figure 8.* Causal graph for the light-tunnel experiment. The confounder $U$ (red) is treated as latent during estimation. The edge $X \to M$ is implemented programmatically by the chamber controller via $M := 0.0625 \cdot X$ between sensor reads.

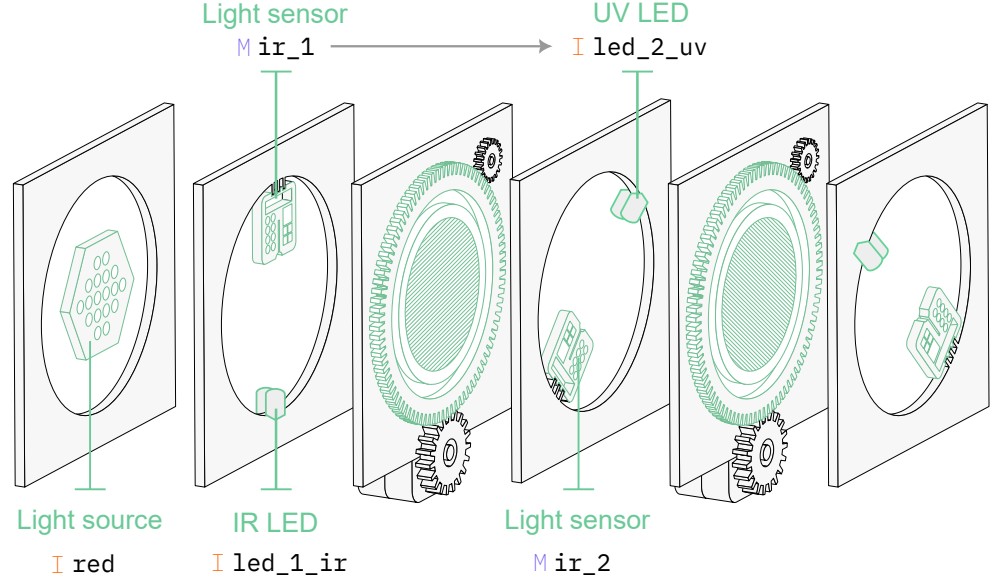

*Figure 9.* Schematic of the Causal Chamber light tunnel. Figure adapted from Gamella et al. (2025), licensed under CC BY 4.0; see also https://cchamber-box.s3.eu-central-2.amazonaws.com/config_doc_lt_mk2_standard.pdf.

**Causal model.** We generate the data according to the DAG in Figure 8. The red-channel intensity $U$ affects both the treatment measurement $X$ and the downstream outcome $Y$ (in this experiment, we assume that $U$ is unobserved). To identify the causal effect of $X$ on $Y$, we use $I = $ led_1_ir as an instrument that shifts $X$ while (by design of the protocol) remaining independent of $U$ and affecting $Y$ only through $X$. The effect of $X$ on $Y$ is not direct in the hardware: it is mediated by $M = $ led_2_uv, which is set by a controller according to a deterministic mapping $M := 0.0625 \cdot X$ executed between the measurement of ir_1 and the measurement of ir_2. This creates a controlled, mechanistic pathway $X \to M \to Y$ while keeping the confounding path $X \leftarrow U \to Y$ intact.

**Protocol and ground truth.** Each observation is collected in two stages: (i) we set $(I, U)$, then measure $X = $ ir_1; (ii) the chamber sets $M = $ led_2_uv using $M := 0.0625 \cdot X$, then we measure $Y = $ ir_2. We sample $U$ uniformly from $0, \dots, 255$ and independently (and uniformly with replacement) choose $I$ from $m$ discrete levels in $0, \dots, 749$ (repeated $r$ times each), yielding $n = mr$ observations. By construction, this dataset is paired. We artificially unpair it by removing either $X$ or $Y$ in each data point.

Although $U$ is treated as unobserved by the estimator, it is recorded by the experimenter; hence we can compute oracle targets by adjustment (on the paired dataset).

The p-value of a Pearson correlation t-test for the one-sided positive correlation between $X$ and $I$ is $6.5 \cdot 10^{-8}$. We compute the difference of $R^2$-values between a model $Y \sim 1 + X + U$ and $Y \sim 1 + X + U + X^2$ to measure the amount of non-linearity. The results ($\Delta R^2 = 0.000172163$) indicated a very weak non-linear effect. We also measure the difference

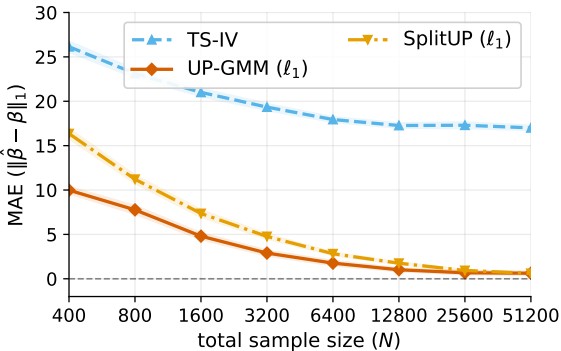

*Figure 10.* **Setting 1 (finite-dimensional instruments, sparse $\beta^*$).** Both UP-GMM and SPLITUP are consistent, whereas the estimation error of TS-IV does not vanish even at large sample sizes. Because this setting is low-dimensional, the bias correction in SPLITUP is unnecessary and because of the increased variance SPLITUP performs worse than UP-GMM, especially for small sample sizes.

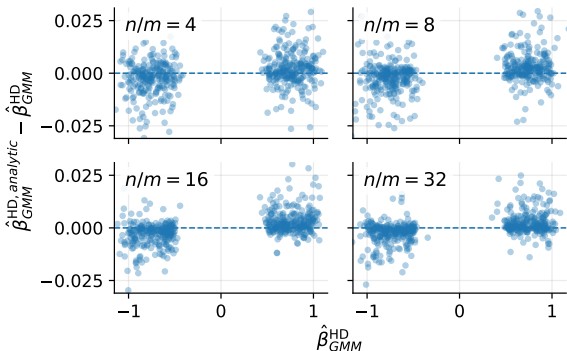

*Figure 11.* **Agreement between SPLITUP and SPLITUP (analytic).** In Setting 2, we compare SPLITUP to SPLITUP (analytic). To mitigate numerical instability when $n/m = 32$ and $N$ is small, we remove the $0.1\%$ of points with the largest discrepancy (outliers). The two estimators agree closely, remaining within a $2.5\%$ margin of each other.

between the regression coefficient of $X$ between the models $Y \sim 1 + X + U$ and $Y \sim 1 + X + U + I$. The result is a $3\%$ difference indicating no (or a very weak) violation of the exclusion condition.

## I. Additional Experiments

### I.1. Setting 1: Finite-Dimensional Categorical Instruments, Sparse $\beta^*$.

Results are shown in Figure 10. In this regime, the moment matrix can be rank-deficient even asymptotically, so the unregularized TS-IV estimator does not leverage sparsity and fails to recover $\beta^*$ reliably. In contrast, both UP-GMM and SPLITUP exploit the sparse structure and empirically approach the oracle target as $n$ increases. For small and moderate sample sizes, UP-GMM typically performs slightly better than SPLITUP: cross-fitting is not needed to remove high-dimensional measurement-error bias here and the splitting in SPLITUP slightly increases variance.

### I.2. Agreement between SPLITUP and SPLITUP (analytic).

Figure 11 show that the estimates of SPLITUP and SPLITUP (analytic) agree closely in Setting 2 (with categorical instruments).

### I.3. Continuous Instrument

The experimental details can be found in Appendix H.2. The results for Setting 1 are given in Figure 12 and the results of Setting 2 and 3 in Figure 13.

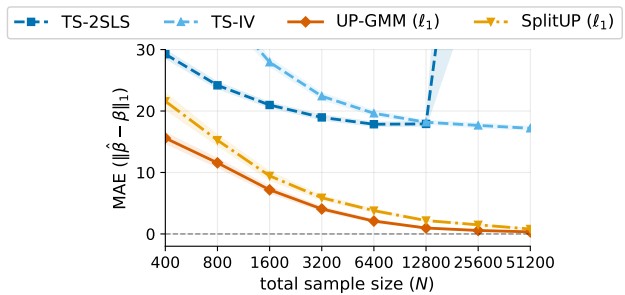

*Figure 12.* **Setting 1 (continuous).** UP-GMM ($\ell_1$-regularized) and SPLITUP ($\ell_1$-regularized) are consistent while TS-IV and TS-2SLS are not.

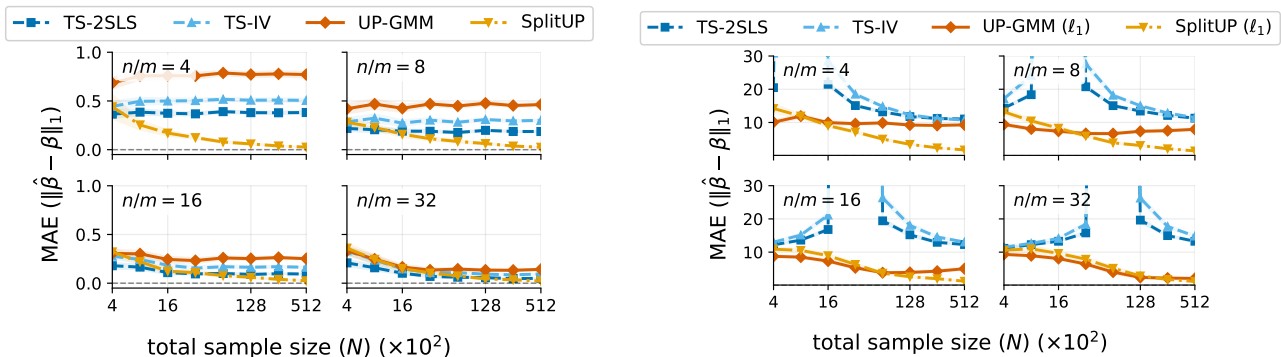

*Figure 13.* **Left: Setting 2 (continuous).** Only SPLITUP is consistent, all other estimators remain biased even for large sample-sizes. The bias reduces as $n/m$ increases and the problem becomes less high-dimensional. **Right: Setting 3 (continuous).** Only SPLITUP ($\ell_1$-regularized) is consistent, all other estimators remain biased even for large sample-sizes. The bias reduces for UP-GMM ($\ell_1$-regularized) as $n/m$ increases and the problem becomes less high-dimensional.

## I.4. High-Dimensional Causal Effect

We Setting 2 (categorical) as described in Appendix H.1. We set $d = 1000$, $\gamma_x = 1$, $\gamma_y = 1$, $\sigma_u = 0.02$, $\sigma_x = 0.02$, $\sigma_\varepsilon = 0.02$, and sample the environment means also with standard deviation 0.02. For sparse UP-GMM, we set $\lambda_1 = 0.0002\alpha_{\max}\sqrt{\log(d+1)/m}$. For sparse SPLITUP, we use $\lambda_1 = 0.0002\sqrt{\log(d+1)/m}$ after diagonal normalization of $C_{XX}$. Here, we hand-tuned $\lambda_1$ (but have not seen, for any $\lambda_1$, UP-GMM outperforming SPLITUP, not shown). For a data-driven procedure to select the $\ell_1$ penalty, see Appendix I.5. The results are shown in Figure 14.

## I.5. Selection of the $\ell_1$ penalty with Bootstrapping

Because the data are unpaired, standard prediction-based cross-validation is not available and, in our setting, would in any case be conceptually misaligned with the role of sparsity. Here sparsity is used mainly to resolve identifiability of the moment equation rather than to optimize held-out prediction. We therefore use a residual-based selection rule motivated by the noiseless population problem: choose the largest $\lambda$ for which the fitted sparse solution is still compatible with zero population residual.

For each $\lambda$ on a decreasing logarithmic grid, we compute

$$\hat{\beta}_\lambda \in \arg\min_\beta \frac{1}{2}\|C_{XY} - C_{XX}\beta\|_2^2 + \lambda\|\beta\|_1$$

and the residual $R(\lambda) = \|C_{XY} - C_{XX}\hat{\beta}_\lambda\|_2$. We then estimate, via multiplier bootstrap, a critical value $\hat{c}_{1-\alpha}(\lambda)$ for the residual fluctuation expected under zero population residual, and set

$$\hat{\lambda} = \max\{\lambda \in \Lambda : R(\lambda) \leqslant \hat{c}_{1-\alpha}(\lambda)\}.$$

This rule is conservative in the intended direction: it favors the sparsest model whose residual can still be explained by

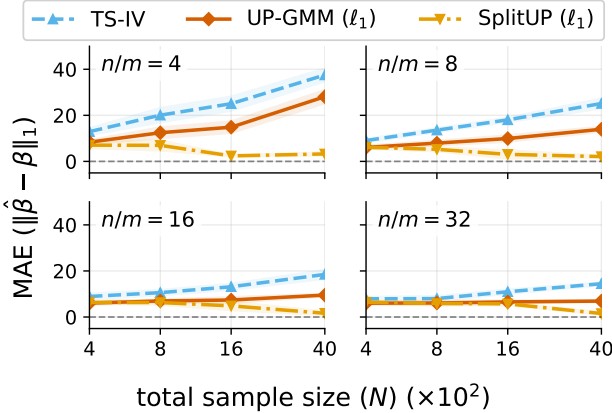

*Figure 14.* **Setting 2 (categorical) with large** $d$**.** SPLITUP ($\ell_1$-regularized) is consistent while UP-GMM ($\ell_1$-regularized) and TS-IV are biased. As in Setting 3 (categorical) we see that the measurement-in-error bias increases with $m$; see also Appendix J. We plot averages over 5 independent runs.

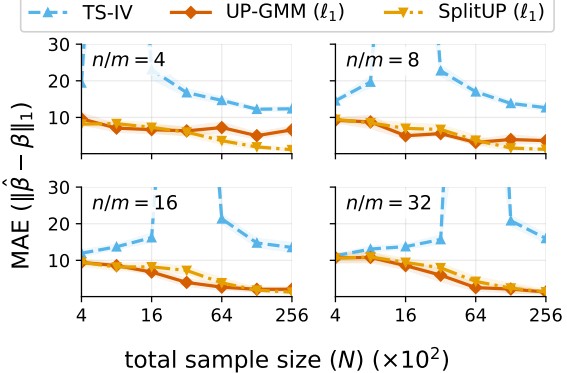

*Figure 15.* **Setting 3 with** $\ell_1$ **selection via bootstrapping.** Results for the "Setting 3" experiment as described in Section 5 but where we use Algorithm 9 to select the $\ell_1$ penalty for SPLITUP and UP-GMM. Note that for computational reasons we show results only upto $N = 25600$ as compared to the corresponding plot in Section 5 where we additionally give results for $n = 51200$. We plot averages over 10 independent runs.

estimation noise. After selecting $\hat{\lambda}$, we refit the unpenalized estimator on the selected support. The ridge parameter is fixed to a very small constant for numerical stabilization and is not tuned. A pseudocode for this method is given in Algorithm 9. We also empirically validate this approach for data generated as in Setting 3 (described in Section 5) where we choose $\ell_1$ penalty for both SPLITUP and UP-GMM with Algorithm 9. The results are shown in Figure 15. We see that performance is similar to the heuristical penalty selection used for experiments in Section 5.

---

**Algorithm 9** Bootstrap selector for $\lambda$

---

1: **Input:** moments $(C_{XX}, C_{XY})$, decreasing grid $\Lambda$, level $\alpha$, bootstrap repetitions $B$
2: **for** $\lambda \in \Lambda$ **do**
3:     Compute $\hat{\beta}_\lambda$ from the $\ell_1$-penalized objective
4:     $R(\lambda) \leftarrow \|C_{XY} - C_{XX}\hat{\beta}_\lambda\|_2$
5:     **for** $b = 1, \ldots, B$ **do**
6:         Generate centered multiplier weights on the $X$-sample and $Y$-sample
7:         Form bootstrap perturbations $\Delta C_{XX}^{(b)}$ and $\Delta C_{XY}^{(b)}$
8:         $T_b(\lambda) \leftarrow \|\Delta C_{XY}^{(b)} - \Delta C_{XX}^{(b)}\hat{\beta}_\lambda\|_2$
9:     **end for**
10:    Let $\hat{c}_{1-\alpha}(\lambda)$ be the $(1 - \alpha)$-quantile of $\{T_b(\lambda)\}_{b=1}^{B}$
11:    **if** $R(\lambda) \leqslant \hat{c}_{1-\alpha}(\lambda)$ **then**
12:       select this $\lambda$ and stop                           (largest accepted $\lambda$)
13:    **end if**
14: **end for**
15: Optional: refit the unpenalized estimator on $\mathrm{supp}(\hat{\beta}_{\hat{\lambda}})$

---

## J. Hypothesized Explanation of the TS-IV Peaking Phenomenon in Setting 3

In Setting 3 we observe a non-monotone error curve for TS-IV as a function of $m$ (a "peaking" phenomenon). A plausible explanation is most transparent from the spectral form of the ridge-stabilized TS-IV estimator

$$\hat{\beta} = (B^\top B + \lambda I)^{-1} B^\top a, \qquad B = C + \Delta,$$

where $C = \mathrm{Cov}(I, X)$ is low-rank by construction (hence $\sigma_{\min}(C) = 0$) and $\Delta$ collects sampling noise from estimating $\mathrm{Cov}(I, X)$. Let $B = U\Sigma V^\top$ with singular values $\sigma_i$. Then

$$\hat{\beta} = V \, \mathrm{diag}\!\left(\frac{\sigma_i}{\sigma_i^2 + \lambda}\right) U^\top a,$$

so each singular direction is weighted by the filter $g(\sigma) := \sigma/(\sigma^2 + \lambda)$. Importantly, nullspace directions are *not* where instability comes from: if $\sigma_i = 0$, then $g(0) = 0$, so those components are simply set to (near) zero (bias but no explosion). Instead, the potential blow-up comes from *near-null* directions with small but nonzero $\sigma_i$, because $g(\sigma)$ is maximized at $\sigma = \sqrt{\lambda}$ with peak value $g(\sqrt{\lambda}) = 1/(2\sqrt{\lambda})$; thus very small ridge $\lambda$ (as in our implementation) can lead to large amplification whenever the smallest *positive* singular values of $B$ approach $\sqrt{\lambda}$.

Random-matrix heuristics suggest that the smallest positive singular value of a noise-like $m \times d$ matrix is controlled by the "rectangularity gap" $|\sqrt{d} - \sqrt{m}|$; see, e.g., equation (2.1) in Rudelson and Vershynin (2010) (up to constants and the appropriate variance normalization). Consequently, near the almost-square regime $m \approx d$, the gap $|\sqrt{d} - \sqrt{m}|$ is small and the smallest positive singular values can be very close to zero, placing TS-IV in the high-gain regime of $g(\sigma)$ and producing a peak. When $m$ is much smaller than $d$, the gap $\sqrt{d} - \sqrt{m}$ is larger, pushing the smallest *positive* singular values away from zero and reducing amplification; moreover, because $C$ is low-rank, aggressively shrinking the many orthogonal (mostly-noise) directions can reduce variance substantially with limited additional bias, which can make performance look comparatively "good" even when $m < d$. Finally, note that in Setting 1 the dimensions $d$ and $m$ are fixed, so this near-square random-matrix mechanism is not present and no peaking is expected. While we emphasize that this is a hypothesis, not a full proof, it seems to be supported by the experiments. If our theory is correct then we would expect the peak when $m \approx d = 100$. Since $N = 2rm$ in our setting we expect the peaks at $N = 200r$. For $r = 4$ this implies $N = 800$ and for $r = 32$ at $N = 6400$. These points are exactly where we see the peaking in Figure 4.

## K. Consistency and Asymptotic CIs for the Dense Setting

For the optimal choice, estimate the variance of the sample moment at a preliminary consistent estimator $\hat{\beta}^{(0)}$ (for instance (4.2) with $W_N = \mathrm{Id}_m$) via

$$\widehat{\Omega}_m := \frac{1}{n} \sum_{i=1}^{n} \left(I_i Y_i - \frac{1}{n}\sum_{r=1}^{n} I_r Y_r\right)\left(I_i Y_i - \frac{1}{n}\sum_{r=1}^{n} I_r Y_r\right)^\top,$$

$$\widehat{\Omega}_c\big(\hat\beta^{(0)}\big) := \frac{1}{\tilde n}\sum_{j=1}^{\tilde n}\Big(\tilde I_j \tilde X_j^\top \hat\beta^{(0)} - \Big[\frac{1}{\tilde n}\sum_{s=1}^{\tilde n}\tilde I_s \tilde X_s^\top\Big]\hat\beta^{(0)}\Big)\Big(\tilde I_j \tilde X_j^\top \hat\beta^{(0)} - \Big[\frac{1}{\tilde n}\sum_{s=1}^{\tilde n}\tilde I_s \tilde X_s^\top\Big]\hat\beta^{(0)}\Big)^\top.$$

Set

$$\widehat\Omega := \tau_n^{-1}\widehat\Omega_m + \tilde\tau_n^{-1}\widehat\Omega_c\big(\hat\beta^{(0)}\big), \qquad \widehat W := \widehat\Omega^{-1}. \tag{K.1}$$

The asymptotic variance of the estimator is minimized by replacing $W_N$ with $\widehat W$ in (4.2). Define

$$V(W_0) := \Big(\operatorname{Cov}(\tilde I,\tilde X)^\top W_0 \operatorname{Cov}(\tilde I,\tilde X)\Big)^{-1}\operatorname{Cov}(\tilde I,\tilde X)^\top W_0 \Omega W_0$$
$$\operatorname{Cov}(\tilde I,\tilde X)\Big(\operatorname{Cov}(\tilde I,\tilde X)^\top W_0 \operatorname{Cov}(\tilde I,\tilde X)\Big)^{-1}.$$

**Assumption 6.** *Assume all of the following:*

  (i) *Assumption 1, the centering convention and with bounded fourth moments.*
 (ii) $\operatorname{rank}\big(\operatorname{Cov}(I,X)\big) = d.$
(iii) $W_N \overset{p}{\to} W_0$ *with $W_0$ positive definite.*

**Proposition K.1** (Consistency and asymptotic normality, dense case; formal version of Proposition 4.1). *Under Assumption 6, the estimator* (4.2) *is consistent:*

$$\hat\beta_{\mathrm{GMM}}^{\mathrm{UP}}(W_N) \overset{p}{\to} \beta^*. \tag{K.2}$$

*Then*

$$\sqrt{N}\big(\hat\beta_{\mathrm{GMM}}^{\mathrm{UP}}(W_N) - \beta^*\big) \overset{d}{\to} \mathcal{N}\Big(0, V(W_0)\Big). \tag{K.3}$$

*With the choice $W_0 = \Omega^{-1}$ the variance in* (K.3) *is minimized in the usual GMM sense. Let $\widehat V$ be the empirical version of $V(\Omega^{-1})$. Then*

$$\sqrt{N}\widehat V^{-1/2}\big(\hat\beta_{\mathrm{GMM}}^{\mathrm{UP}}(W_N) - \beta^*\big) \overset{d}{\to} \mathcal{N}\Big(0, \mathrm{Id}_d\Big).$$

*Proof.* By the strong law,

$$\frac{1}{n}\sum_{i=1}^{n} I_i Y_i \overset{p}{\to} \mathbb{E}[IY] = \operatorname{Cov}(I,Y), \qquad \frac{1}{\tilde n}\sum_{j=1}^{\tilde n}\tilde I_j \tilde X_j^\top \overset{p}{\to} \mathbb{E}[\tilde I \tilde X^\top] = \operatorname{Cov}(\tilde I,\tilde X).$$

Therefore $\hat g_N(\beta) \overset{p}{\to} \operatorname{Cov}(I,Y) - \operatorname{Cov}(\tilde I,\tilde X)\beta$. The continuous mapping theorem and Assumption 6 (ii) imply that the population criterion $Q(\beta) = \big(\operatorname{Cov}(I,Y) - \operatorname{Cov}(\tilde I,\tilde X)\beta\big)^\top W_0\big(\operatorname{Cov}(I,Y) - \operatorname{Cov}(\tilde I,\tilde X)\beta\big)$ has the unique minimizer $\beta^*$. Uniform convergence of the quadratic sample criterion to $Q$ yields consistency of (4.2).

For asymptotic normality, write the first–order condition

$$\Big(\frac{1}{\tilde n}\sum_{j=1}^{\tilde n}\tilde X_j \tilde I_j^\top\Big)W_N \hat g_N(\hat\beta_{\mathrm{GMM}}^{\mathrm{UP}}(W_N)) = 0.$$

Add and subtract $\beta^*$ inside $\hat g_N$ and rearrange:

$$\Big(\frac{1}{\tilde n}\sum_{j=1}^{\tilde n}\tilde X_j \tilde I_j^\top\Big)W_N\Big(\frac{1}{\tilde n}\sum_{j=1}^{\tilde n}\tilde I_j \tilde X_j^\top\Big)\big(\hat\beta_{\mathrm{GMM}}^{\mathrm{UP}}(W_N) - \beta^*\big) = \Big(\frac{1}{\tilde n}\sum_{j=1}^{\tilde n}\tilde X_j \tilde I_j^\top\Big)W_N \hat g_N(\beta^*).$$

By Slutsky, the left matrix converges in probability to $\operatorname{Cov}(\tilde I,\tilde X)^\top W_0 \operatorname{Cov}(\tilde I,\tilde X)$, which is invertible by Assumption 6(ii). Multiplying both sides by its inverse and by $\sqrt{N}$ yields

$$\sqrt{N}\big(\hat\beta_{\mathrm{GMM}}^{\mathrm{UP}}(W_N) - \beta^*\big) = \Big(\operatorname{Cov}(\tilde I,\tilde X)^\top W_0 \operatorname{Cov}(\tilde I,\tilde X)\Big)^{-1}\operatorname{Cov}(\tilde I,\tilde X)^\top W_0 \sqrt{N}\hat g_N(\beta^*) + o_p(1). \tag{K.4}$$

Finally, apply a joint CLT to the two independent samples:

$$\sqrt{N}\hat g_N(\beta^*) = \sqrt{N}\Big(\frac{1}{n}\sum_{i=1}^{n}(I_i Y_i - \mathbb{E}[IY])\Big) - \sqrt{N}\Big(\frac{1}{\tilde n}\sum_{j=1}^{\tilde n}(\tilde I_j \tilde X_j^\top - \mathbb{E}[\tilde I \tilde X^\top])\Big)\beta^*$$
$$\overset{d}{\to} \mathcal{N}\big(0, \tau^{-1}\Omega_m + \tilde\tau^{-1}\Omega_c(\beta^*)\big), \tag{K.5}$$

because $\sqrt{N/n} \to \tau^{-1/2}$ and $\sqrt{N/\tilde n} \to \tilde\tau^{-1/2}$. Plug (K.5) into (K.4) to obtain (K.3). $\qquad\square$

# L. Proofs

## L.1. Proof of Proposition 3.1

If $\operatorname{rank}(\operatorname{Cov}(I, X)) = d$, then $\operatorname{Cov}(I, X)$ (and therefore $\operatorname{Cov}(\tilde{I}, \tilde{X})$) has a left inverse and there exists a unique solution $\hat{\beta} = \beta^*$ to

$$\operatorname{Cov}(I, Y) = \operatorname{Cov}(\tilde{I}, \tilde{X})\beta$$

given by

$$\hat{\beta} = \left(\operatorname{Cov}(\tilde{I}, \tilde{X})^\top \operatorname{Cov}(\tilde{I}, \tilde{X})\right)^{-1} \operatorname{Cov}(\tilde{I}, \tilde{X})^\top \operatorname{Cov}(I, Y) = \beta^*.$$

If

$$\operatorname{rank}\left(\operatorname{cov}(\tilde{I}, \tilde{X})\right) = \operatorname{rank}\left(\operatorname{cov}(I, X)\right) < d,$$

then

$$\ker\left(\operatorname{cov}(\tilde{I}, \tilde{X})\right) \neq \{0\},$$

and for any $h$ in the kernel, $\beta^* + h$ yields the same moment, so $\beta^*$ is not identified.

## L.2. Proof of Theorem 3.2

**(i) $\Rightarrow$ (ii).** Suppose (ii) fails. Then there exists $0 \neq h \in \ker(\operatorname{Cov}(I, X)) \cap \Sigma_{2s^*}$. Partition $\operatorname{supp}(h)$ into disjoint $J_1, J_2$ with $|J_i| \leqslant s^*$ and define $u, v \in \Sigma_{s^*}$ by $u_{J_1} = h_{J_1}$, $v_{J_2} = -h_{J_2}$, zeros elsewhere. Then $\operatorname{Cov}(I, X)u = \operatorname{Cov}(I, X)v$ and $\|u\|_0, \|v\|_0 \leqslant s^*$, but $u \neq v$, contradicting (i) for $\beta^* = v$. Hence, (ii) must hold.

**(ii) $\Rightarrow$ (i)** If $\operatorname{Cov}(I, X)\hat{\beta} = \operatorname{Cov}(I, X)\beta^*$ and $\|\hat{\beta}\|_0 \leqslant \|\beta^*\|_0 \leqslant s^*$, then $h := \hat{\beta} - \beta^* \in \ker(\operatorname{Cov}(I, X)) \cap \Sigma_{2s^*}$; by (ii), $h = 0$, so $\hat{\beta} = \beta^*$.

## L.3. Proof of Theorem 3.3

If $\operatorname{rank}(Q) = d$, then there exists a unique solution $\hat{\beta} = \beta^*$ to

$$Q_y = Q\beta$$

given by

$$\hat{\beta} = Q^{-1}Q_y = \beta^*.$$

If $\operatorname{rank}(Q) < d$, then $\ker(Q) \neq \{0\}$, and for any $h$ in the kernel, $\beta^* + h$ yields the same moment, so $\beta^*$ is not identifiable.

## L.4. Proof of Theorem 3.4

**(i) $\Rightarrow$ (ii).** Suppose (ii) fails. Then there exists $0 \neq h \in \ker(Q) \cap \Sigma_{2s^*}$. Partition $\operatorname{supp}(h)$ into disjoint $J_1, J_2$ with $|J_i| \leqslant s^*$ and define $u, v \in \Sigma_{s^*}$ by $u_{J_1} = h_{J_1}$, $v_{J_2} = -h_{J_2}$, zeros elsewhere. Then $Qu = Qv$ and $\|u\|_0, \|v\|_0 \leqslant s^*$, but $u \neq v$, contradicting (i) for $\beta^* = v$. Hence, (ii) must hold.

**(ii) $\Rightarrow$ (i)** If $Q\hat{\beta} = Q\beta^*$ and $\|\hat{\beta}\|_0 \leqslant \|\beta^*\|_0 \leqslant s^*$, then $h := \hat{\beta} - \beta^* \in \ker(Q) \cap \Sigma_{2s^*}$; by (ii), $h = 0$, so $\hat{\beta} = \beta^*$.

## L.5. Proof of Theorem 4.2

**Lemma L.1.** *Let $\hat{g}_N(\beta) := \frac{1}{n} \sum_{i=1}^{n} I_i Y_i - \left(\frac{1}{\tilde{n}} \sum_{j=1}^{\tilde{n}} \tilde{I}_j \tilde{X}_j^\top\right)\beta$ and $\widehat{\operatorname{Cov}}(\tilde{I}, \tilde{X}) := \frac{1}{\tilde{n}} \sum_{j=1}^{\tilde{n}} \tilde{I}_j \tilde{X}_j^\top$. Under Assumption 3,*

$$\left\|\widehat{\operatorname{Cov}}(\tilde{I}, \tilde{X})^\top W_N \hat{g}_N(\beta^*)\right\|_\infty = O_p(N^{-1/2}).$$

*Proof.* By the moment condition $\operatorname{Cov}(I, Y) = \operatorname{Cov}(\tilde{I}, \tilde{X})\beta^*$, we have

$$\hat{g}_N(\beta^*) = \widehat{\operatorname{Cov}}(I, Y) - \widehat{\operatorname{Cov}}(\tilde{I}, \tilde{X})\beta^* = (\widehat{\operatorname{Cov}}(I, Y) - \operatorname{Cov}(I, Y)) - (\widehat{\operatorname{Cov}}(\tilde{I}, \tilde{X}) - \operatorname{Cov}(I, X))\beta^*.$$

Since $m$ is fixed and fourth moments are bounded, Chebyshev (or a multivariate CLT) yields

$$\|\widehat{\mathrm{Cov}}(I, Y) - \mathrm{Cov}(I, Y)\|_2 = O_p(n^{-1/2}), \qquad \|\widehat{\mathrm{Cov}}(\tilde{I}, \tilde{X}) - \mathrm{Cov}(I, X)\|_{\mathrm{op}} = O_p(\tilde{n}^{-1/2}).$$

Using $\|\beta^*\|_2 = O(1)$ and $\tau_n, \tilde{\tau}_n \to (0, 1)$ gives

$$\|\hat{g}_N(\beta^*)\|_2 \leqslant \|\widehat{\mathrm{Cov}}(I, Y) - \mathrm{Cov}(I, Y)\|_2 + \|\widehat{\mathrm{Cov}}(\tilde{I}, \tilde{X}) - \mathrm{Cov}(I, X)\|_{\mathrm{op}}\|\beta^*\|_2 = O_p(n^{-1/2} + \tilde{n}^{-1/2}) = O_p(N^{-1/2}).$$

Moreover, by the law of large numbers in fixed dimensions, $\|\widehat{\mathrm{Cov}}(\tilde{I}, \tilde{X})\|_{\mathrm{op}} = O_p(1)$, and by $W_N \xrightarrow{p} W_0 \succ 0$ we have $\|W_N\|_{\mathrm{op}} = O_p(1)$. Therefore,

$$\|\widehat{\mathrm{Cov}}(\tilde{I}, \tilde{X})^\top W_N \hat{g}_N(\beta^*)\|_\infty \leqslant \|\widehat{\mathrm{Cov}}(\tilde{I}, \tilde{X})^\top W_N \hat{g}_N(\beta^*)\|_2 \leqslant \|\widehat{\mathrm{Cov}}(\tilde{I}, \tilde{X})\|_{\mathrm{op}}\|W_N\|_{\mathrm{op}}\|\hat{g}_N(\beta^*)\|_2 = O_p(N^{-1/2}).$$

$\square$

Let

$$S := \mathrm{supp}(\beta^*), \qquad s^* := |S|.$$

The optimality of (4.4) yields the basic inequality

$$\frac{1}{2}\left\|W_N^{1/2}\hat{g}_N(\hat{\beta}^{\mathrm{UP}}_{\mathrm{GMM},\ell_1})\right\|_2^2 + \lambda_N\|\hat{\beta}^{\mathrm{UP}}_{\mathrm{GMM},\ell_1}\|_1 \leqslant \frac{1}{2}\left\|W_N^{1/2}\hat{g}_N(\beta^*)\right\|_2^2 + \lambda_N\|\beta^*\|_1. \tag{L.1}$$

Using the linearization

$$\hat{g}_N(\hat{\beta}^{\mathrm{UP}}_{\mathrm{GMM},\ell_1}) = \hat{g}_N(\beta^*) - \left(\frac{1}{\tilde{n}}\sum_{j=1}^{\tilde{n}}\tilde{I}_j\tilde{X}_j^\top\right)(\hat{\beta}^{\mathrm{UP}}_{\mathrm{GMM},\ell_1} - \beta^*)$$

and expanding (L.1), one obtains

$$\left(\hat{\beta}^{\mathrm{UP}}_{\mathrm{GMM},\ell_1} - \beta^*\right)^\top\left(\frac{1}{\tilde{n}}\sum_{j=1}^{\tilde{n}}\tilde{X}_j\tilde{I}_j^\top\right)W_N\left(\frac{1}{\tilde{n}}\sum_{j=1}^{\tilde{n}}\tilde{I}_j\tilde{X}_j^\top\right)(\hat{\beta}^{\mathrm{UP}}_{\mathrm{GMM},\ell_1} - \beta^*) \tag{L.2}$$

$$\leqslant 2\lambda_N\left(\|\beta^*\|_1 - \|\hat{\beta}^{\mathrm{UP}}_{\mathrm{GMM},\ell_1}\|_1\right) + 2\left\|\frac{1}{\tilde{n}}\sum_{j=1}^{\tilde{n}}\tilde{X}_j\tilde{I}_j^\top W_N\hat{g}_N(\beta^*)\right\|_\infty\|\hat{\beta}^{\mathrm{UP}}_{\mathrm{GMM},\ell_1} - \beta^*\|_1. \tag{L.3}$$

By Lemma L.1

$$\left\|\frac{1}{\tilde{n}}\sum_{j=1}^{\tilde{n}}\tilde{X}_j\tilde{I}_j^\top W_N\hat{g}_N(\beta^*)\right\|_\infty = O_p\left(\sqrt{\frac{1}{N}}\right). \tag{L.4}$$

Choose the constant in $\lambda_N$ so that the right side of (L.4) is bounded by

$$\frac{\lambda_N}{2}$$

with high probability. Then the usual cone condition

$$\|\Delta_{S^c}\|_1 \leqslant 3\|\Delta_S\|_1$$

holds for

$$\Delta := \hat{\beta}^{\mathrm{UP}}_{\mathrm{GMM},\ell_1} - \beta^*.$$

By Assumption 3 (i), Assumption 3 (ii) and Slutsky's theorem we have

$$\widehat{\mathrm{Cov}}(\tilde{I}, \tilde{X})^\top W_N \widehat{\mathrm{Cov}}(\tilde{I}, \tilde{X}) \xrightarrow{p} \mathrm{Cov}(\tilde{I}, \tilde{X})^\top W_0 \mathrm{Cov}(\tilde{I}, \tilde{X})$$

(point-wise and therefore also in operator norm). Therefore, by Assumption 3 (iii), we have, for large enough $N$ on the cone $\|\Delta_{S^c}\|_1 \leqslant 3\|\Delta_S\|_1$, that

$$
\begin{aligned}
\kappa\|\Delta\|_2^2/2 &\leqslant \Delta^\top \Big(\frac{1}{\tilde{n}}\sum_{j=1}^{\tilde{n}} \tilde{X}_j \tilde{I}_j^\top\Big) W_N \Big(\frac{1}{\tilde{n}}\sum_{j=1}^{\tilde{n}} \tilde{I}_j \tilde{X}_j^\top\Big)\Delta \\
&\leqslant 3\lambda_N\|\Delta\|_1 \\
&\leqslant 3\lambda_N\sqrt{s^*}\|\Delta\|_2,
\end{aligned}
\tag{L.5}
$$

which gives

$$
\|\Delta\|_2 \in O_p\left(\sqrt{\frac{s^*}{N}}\right)
$$

and then

$$
\|\Delta\|_1 \in O_p\left(s^*\sqrt{\frac{1}{N}}\right).
$$

### L.6. Proof of Theorem E.1

By Theorem 4.2 we have $\mathbb{P}(\hat{S}(W_N) = S^*) \to 1$. On the event $\{\hat{S} = S^*\}$, the refit equals the oracle GMM estimator on $S^*$. Standard GMM theory (see Appendix K and note that Assumption 3 (iii) implies Assumption 6 (ii) on $S^*$, i.e. ,we have $\text{rank}(\text{Cov}(I, X)_{S^*}) = s^*$) yields $\sqrt{N}(\tilde{\beta}_{S^*}(W_N') - \beta_{S^*}^*) \xrightarrow{d} \mathcal{N}(0, V_{S^*})$ and consistency of the sandwich $\widehat{V}_{\hat{S}}$. Slutsky's lemma and $\mathbb{P}(\hat{S} = S^*) \to 1$ transfer the oracle limit to the random-support estimator.

### L.7. Proof of Lemma 4.3

Write

$$
c := \text{Cov}(I, X) \in \mathbb{R}^{m \times d}, \qquad s := \text{Cov}(I, Y) \in \mathbb{R}^m, \qquad \bar{c} := \sqrt{m}c, \qquad \bar{s} := \sqrt{m}s.
$$

Define

$$
E_k := \sqrt{m}\widehat{\text{Cov}}_k(\tilde{I}, \tilde{X}) - \bar{c} = \sqrt{m}\Big(\widehat{\text{Cov}}_k(\tilde{I}, \tilde{X}) - c\Big), \qquad k \in [K].
$$

Define

$$
\hat{c} := \sqrt{m}\widehat{\text{Cov}}(\tilde{I}, \tilde{X}), \qquad E := \hat{c} - \bar{c} = \sqrt{m}\Big(\widehat{\text{Cov}}(\tilde{I}, \tilde{X}) - c\Big).
$$

For the $(I, Y)$ sample define

$$
\hat{s} := \sqrt{m}\widehat{\text{Cov}}(I, Y), \qquad H := \hat{s} - \bar{s} = \sqrt{m}\Big(\widehat{\text{Cov}}(I, Y) - s\Big).
$$

We have

$$
\begin{aligned}
\hat{\beta} &:= \frac{\widehat{\text{Cov}}(\tilde{I}, \tilde{X})^\top \widehat{\text{Cov}}(I, Y)}{\widehat{\text{Cov}}(\tilde{I}, \tilde{X})^\top \widehat{\text{Cov}}(\tilde{I}, \tilde{X})} \\
&= \frac{\bar{c}^\top \bar{s} + \bar{c}^\top H + E^\top \bar{s} + E^\top H}{\bar{c}^\top \bar{c} + \bar{c}^\top E + E^\top \bar{c} + E^\top E}.
\end{aligned}
$$

By Lemma L.2 we have that $\bar{c}^\top H + E^\top \bar{s} + E^\top H = o_p(1)$ and that $\bar{c}^\top E + E^\top \bar{c} = o_p(1)$ and $E^\top E = b/\tilde{r} + o_p(1)$. Furthermore, it holds that $\bar{s} = \bar{c}\beta^*$. Therefore, $\hat{\beta} \to \beta^* \frac{Q}{Q + b/\tilde{r}}$ in probability.

### L.8. Proof of Theorem 4.5

Write

$$
c := \text{Cov}(I, X) \in \mathbb{R}^{m \times d}, \qquad s := \text{Cov}(I, Y) \in \mathbb{R}^m, \qquad \bar{c} := \sqrt{m}c, \qquad \bar{s} := \sqrt{m}s.
$$

Define

$$
\hat{c}_k := \sqrt{m}\widehat{\text{Cov}}_k(\tilde{I}, \tilde{X}), \qquad E_k := \hat{c}_k - \bar{c} = \sqrt{m}\Big(\widehat{\text{Cov}}_k(\tilde{I}, \tilde{X}) - c\Big), \qquad k \in [K].
$$

Define

$$\hat{c} := \sqrt{m}\widehat{\mathrm{Cov}}(\tilde{I}, \tilde{X}), \qquad E := \hat{c} - \bar{c} = \sqrt{m}\left(\widehat{\mathrm{Cov}}(\tilde{I}, \tilde{X}) - c\right).$$

For the $(I, Y)$ sample define

$$\hat{s} := \sqrt{m}\widehat{\mathrm{Cov}}(I, Y), \qquad H := \hat{s} - \bar{s} = \sqrt{m}\left(\widehat{\mathrm{Cov}}(I, Y) - s\right).$$

**Lemma L.2.** *Assume the setting of Theorem 4.5. Let $d \in \mathbb{N}$ and $K \geqslant 2$ be fixed, and $n/m \to r \in (0, \infty)$, $\tilde{n}/m \to \tilde{r} \in (0, \infty)$. Then as $m \to \infty$:*

*(i) For each fixed $k \in [K]$,*

$$\|\bar{c}^\top E_k\|_{\mathrm{op}} = O_p(m^{-1/2}), \qquad \|\bar{c}^\top E\|_{\mathrm{op}} = O_p(m^{-1/2}).$$

*(ii) For $h \neq k$,*

$$\|E_h^\top E_k\|_{\mathrm{op}} = O_p(m^{-1/2}).$$

*(iii) Cross-sample terms vanish:*

$$\|E_k^\top H\|_2 = O_p(m^{-1/2}), \qquad \|E^\top H\|_2 = O_p(m^{-1/2}), \qquad \|\bar{c}^\top H\|_2 = O_p(m^{-1/2}).$$

*(iv) If additionally $d = 1$, $\mathrm{tr}(\Sigma_{IX}) \to b \in (0, \infty)$ and $\sup_m \mathbb{E}[\|IX - c\|_2^4] \leqslant \infty$, then*

$$\|E\|_2^2 \xrightarrow{P} \frac{b}{\tilde{r}}.$$

*Proof.* Throughout, $d$ and $K$ are fixed and $\tilde{n}_k = \tilde{n}/K \asymp m$. By construction, $\widehat{\mathrm{Cov}}_k(\tilde{I}, \tilde{X})$ is the average of $\tilde{n}_k$ i.i.d. copies of $\tilde{I}\tilde{X}$, hence

$$\mathbb{E}[E_k] = 0, \qquad \mathrm{Var}(\mathrm{vec}(E_k)) = \frac{m}{\tilde{n}_k}\Sigma_{IX}.$$

Similarly,

$$\mathbb{E}[E] = 0, \qquad \mathrm{Var}(\mathrm{vec}(E)) = \frac{m}{\tilde{n}}\Sigma_{IX}.$$

Moreover, since $\widehat{\mathrm{Cov}}(I, Y)$ averages $n$ i.i.d. copies of $IY$,

$$\mathbb{E}[H] = 0, \qquad \mathrm{Var}(H) = \frac{m}{n}\Sigma_{IY}.$$

Finally, under Assumption 4 (i) we have $s = c\beta^*$ and thus $\bar{s} = \bar{c}\beta^*$.

**(i).** Fix unit vectors $u, v \in \mathbb{R}^d$. Then

$$v^\top \bar{c}^\top E_k u = (u \otimes \bar{c}v)^\top \mathrm{vec}(E_k),$$

and therefore

$$\begin{aligned}
\mathrm{Var}\left(v^\top \bar{c}^\top E_k u\right) &= (u \otimes \bar{c}v)^\top \mathrm{Var}(\mathrm{vec}(E_k))(u \otimes \bar{c}v) \\
&\leqslant \|\mathrm{Var}(\mathrm{vec}(E_k))\|_{\mathrm{op}}\|u \otimes \bar{c}v\|_2^2 \\
&= \frac{m}{\tilde{n}_k}\|\Sigma_{IX}\|_{\mathrm{op}}\|\bar{c}v\|_2^2 \\
&\leqslant \frac{m}{\tilde{n}_k}\frac{C}{m}\|\bar{c}\|_{\mathrm{op}}^2 = O(m^{-1}),
\end{aligned}$$

where we used $m\|\Sigma_{IX}\|_{\mathrm{op}} \leqslant C$ and $\tilde{n}_k \asymp m$. Hence $v^\top \bar{c}^\top E_k u = O_p(m^{-1/2})$ for each fixed $u, v$. Since $d$ is fixed, $\|\bar{c}^\top E_k\|_{\mathrm{op}} = O_p(m^{-1/2})$. The same argument yields $\|\bar{c}^\top E\|_{\mathrm{op}} = O_p(m^{-1/2})$.

**(ii).** Fix unit vectors $u, v \in \mathbb{R}^d$ and $h \neq k$. Because folds are independent, $\mathbb{E}[u^\top (E_h^\top E_k)v] = 0$ and

$$u^\top (E_h^\top E_k)v = (E_h u)^\top (E_k v).$$

For independent mean-zero vectors $A, B \in \mathbb{R}^m$, $\mathrm{Var}(A^\top B) = \mathrm{tr}(\mathrm{Var}(A)\,\mathrm{Var}(B))$, so

$$\mathrm{Var}\left(u^\top (E_h^\top E_k)v\right) = \mathrm{tr}\left(\mathrm{Var}(E_h u)\,\mathrm{Var}(E_k v)\right) \leqslant m\|\mathrm{Var}(E_h u)\|_{\mathrm{op}}\|\mathrm{Var}(E_k v)\|_{\mathrm{op}}.$$

Now $\mathrm{Var}(\mathrm{vec}(E_h)) = \frac{m}{\tilde{n}_h}\Sigma_{IX}$ implies $\|\mathrm{Var}(E_h u)\|_{\mathrm{op}} \leqslant \frac{m}{\tilde{n}_h}\|\Sigma_{IX}\|_{\mathrm{op}}$ and similarly for $E_k v$, hence

$$\mathrm{Var}\left(u^\top (E_h^\top E_k)v\right) \leqslant m\left(\frac{m}{\tilde{n}_h}\|\Sigma_{IX}\|_{\mathrm{op}}\right)\left(\frac{m}{\tilde{n}_k}\|\Sigma_{IX}\|_{\mathrm{op}}\right) \leqslant m\left(\frac{m}{\Theta(m)}\frac{C}{m}\right)^2 = O(m^{-1}).$$

Therefore $u^\top (E_h^\top E_k)v = O_p(m^{-1/2})$ for each fixed $u, v$, and since $d$ is fixed this yields $\|E_h^\top E_k\|_{\mathrm{op}} = O_p(m^{-1/2})$.

**(iii).** We show $\|E^\top H\|_2 = O_p(m^{-1/2})$; the other statements are analogous. Fix a unit vector $u \in \mathbb{R}^d$. By independence of the $(\tilde{I}, \tilde{X})$ and $(I, Y)$ samples, $\mathbb{E}[u^\top E^\top H] = 0$ and conditioning on $E$ gives

$$\mathrm{Var}(u^\top E^\top H \mid E) = u^\top E^\top \mathrm{Var}(H)Eu = u^\top E^\top \left(\frac{m}{n}\Sigma_{IY}\right)Eu$$

$$\leqslant \frac{m}{n}\|\Sigma_{IY}\|_{\mathrm{op}}\|Eu\|_2^2 \leqslant \frac{m}{n}\frac{C}{m}\|E\|_{\mathrm{op}}^2.$$

Taking expectation and using $\mathbb{E}\|E\|_{\mathrm{op}}^2 \leqslant \mathbb{E}\|E\|_F^2 = \mathrm{tr}(\mathrm{Var}(\mathrm{vec}(E))) = \frac{m}{\tilde{n}}\mathrm{tr}(\Sigma_{IX}) \leqslant \frac{m}{\tilde{n}}\cdot md\|\Sigma_{IX}\|_{\mathrm{op}} \leqslant \frac{m}{\tilde{n}}\cdot md \cdot \frac{C}{m} = O(1)$ yields

$$\mathrm{Var}(u^\top E^\top H) = O(m^{-1}).$$

Thus $u^\top E^\top H = O_p(m^{-1/2})$ for each fixed $u$, and since $d$ is fixed, $\|E^\top H\|_2 = O_p(m^{-1/2})$. The bound for $\|E_k^\top H\|_2$ follows by the same argument with $\tilde{n}$ replaced by $\tilde{n}_k$. Finally, $\|\bar{c}^\top H\|_2 = O_p(m^{-1/2})$ follows from

$$\mathrm{Var}(\bar{c}^\top H) = \bar{c}^\top \mathrm{Var}(H)\bar{c} = \bar{c}^\top \left(\frac{m}{n}\Sigma_{IY}\right)\bar{c} \leqslant \frac{m}{n}\|\Sigma_{IY}\|_{\mathrm{op}}\|\bar{c}\|_F^2 \leqslant \frac{m}{\Theta(m)}\frac{C}{m}\|\bar{c}\|_F^2 = O(m^{-1}),$$

using $\|\bar{c}\|_2^2 = m\|c\|_2^2 = \mathrm{tr}(mc^\top c) = O(1)$.

**(iv).** Assume $d = 1$ and write $Z_j := IX - c \in \mathbb{R}^m$ for the centered summands in the first-stage covariance. Then

$$E = \sqrt{m}\left(\frac{1}{\tilde{n}}\sum_{j=1}^{\tilde{n}} Z_j\right), \qquad \|E\|_2^2 = \frac{m}{\tilde{n}^2}\sum_{j=1}^{\tilde{n}} \|Z_j\|_2^2 + \frac{m}{\tilde{n}^2}\sum_{i\neq j} Z_i^\top Z_j =: A_m + B_m.$$

We have

$$\mathbb{E}[A_m] = \frac{m}{\tilde{n}}\mathbb{E}\|Z_1\|_2^2 = \frac{m}{\tilde{n}}\mathrm{tr}(\Sigma_{IX}) \to \frac{b}{\tilde{r}}.$$

Moreover, under uniformly bounded fourth moments ($\sup_m \mathbb{E}[\|IX - c\|_2^4] \leqslant \infty$) of $\|Z_1\|_2$, $\mathrm{Var}(A_m) = O(1/m)$, hence $A_m - \mathbb{E}[A_m] = o_p(1)$.

Next, $\mathbb{E}[B_m] = 0$ by independence and centering. For the variance of $B_m$, write

$$S := \sum_{i\neq j} Z_i^\top Z_j = 2\sum_{1\leqslant i<j\leqslant \tilde{n}} Z_i^\top Z_j.$$

We claim that the summands $Z_i^\top Z_j$ are pairwise uncorrelated across distinct unordered pairs. Indeed, if $(i, j) \neq (k, \ell)$ as unordered pairs, then either the pairs are disjoint, in which case independence gives zero covariance, or they share exactly one index, say $i = k$ and $j \neq \ell$, in which case

$$\mathbb{E}[(Z_i^\top Z_j)(Z_i^\top Z_\ell)] = \mathbb{E}\left[Z_i^\top \mathbb{E}[Z_j]Z_i^\top \mathbb{E}[Z_\ell]\right] = 0$$

since $Z_j, Z_\ell$ are independent of $Z_i$ and mean zero. Hence

$$\text{Var}\left(\sum_{i<j} Z_i^\top Z_j\right) = \binom{\tilde{n}}{2} \text{Var}(Z_1^\top Z_2).$$

Moreover, by independence and centering,

$$\text{Var}(Z_1^\top Z_2) = \mathbb{E}[(Z_1^\top Z_2)^2] = \text{tr}(\Sigma_{IX}^2).$$

Therefore

$$\text{Var}(S) = 4\binom{\tilde{n}}{2} \text{tr}(\Sigma_{IX}^2) = 2\tilde{n}(\tilde{n}-1)\text{tr}(\Sigma_{IX}^2),$$

and thus

$$\text{Var}(B_m) = \left(\frac{m}{\tilde{n}^2}\right)^2 \text{Var}(S) = 2\frac{m^2(\tilde{n}-1)}{\tilde{n}^3}\text{tr}(\Sigma_{IX}^2) = O\left(\frac{m^2}{\tilde{n}^2}\text{tr}(\Sigma_{IX}^2)\right).$$

Since $\tilde{n} \asymp m$, this is $O(\text{tr}(\Sigma_{IX}^2))$.

Since $\text{tr}(\Sigma_{IX}^2) \leqslant md\|\Sigma_{IX}\|_{\text{op}}^2 \leqslant md(C/m)^2 = O(1/m) \to 0$, we get $B_m = o_p(1)$. Combining yields $\|E\|_2^2 = A_m + B_m \xrightarrow{p} b/\tilde{r}$. $\qquad\square$

Recall the normalized cross-moment definitions

$$C_{XX} := \frac{m}{K(K-1)} \sum_{h \neq k} \widehat{\text{Cov}}_h(\tilde{I}, \tilde{X})^\top \widehat{\text{Cov}}_k(\tilde{I}, \tilde{X}), \qquad C_{XY} := m\widehat{\text{Cov}}(\tilde{I}, \tilde{X})^\top \widehat{\text{Cov}}(I, Y).$$

Equivalently, with $\hat{c}_k = \sqrt{m}\widehat{\text{Cov}}_k(\tilde{I}, \tilde{X})$, $\hat{c} = \sqrt{m}\widehat{\text{Cov}}(\tilde{I}, \tilde{X})$ and $\hat{s} = \sqrt{m}\widehat{\text{Cov}}(I, Y)$,

$$C_{XX} = \frac{1}{K(K-1)}\sum_{h \neq k} \hat{c}_h^\top \hat{c}_k, \qquad C_{XY} = \hat{c}^\top \hat{s}.$$

Let $\bar{c} = \sqrt{m}c$ and $\bar{s} = \sqrt{m}s$. Under Assumption 1, $s = c\beta^*$ and thus $\bar{s} = \bar{c}\beta^*$. Write $\hat{c}_k = \bar{c} + E_k$, $\hat{c} = \bar{c} + E$, and $\hat{s} = \bar{s} + H$ as in Lemma L.2.

Expand

$$C_{XX} = \frac{1}{K(K-1)}\sum_{h \neq k}(\bar{c}+E_h)^\top(\bar{c}+E_k)$$

$$= \bar{c}^\top\bar{c} + \frac{1}{K(K-1)}\sum_{h\neq k}\bar{c}^\top E_k + \frac{1}{K(K-1)}\sum_{h\neq k}E_h^\top\bar{c} + \frac{1}{K(K-1)}\sum_{h\neq k}E_h^\top E_k.$$

By Lemma L.2(i) and (ii), each of the last three terms is $o_p(1)$ since $K$ is fixed. Therefore

$$C_{XX} = \bar{c}^\top\bar{c} + o_p(1) = mc^\top c + o_p(1) \xrightarrow{p} Q.$$

Expand

$$C_{XY} = (\bar{c}+E)^\top(\bar{s}+H) = \bar{c}^\top\bar{s} + \bar{c}^\top H + E^\top\bar{s} + E^\top H.$$

Since $\bar{s} = \bar{c}\beta^*$,

$$\bar{c}^\top\bar{s} = \bar{c}^\top\bar{c}\beta^* = mc^\top c\beta^* \to Q\beta^*.$$

Moreover, Lemma L.2(iii) gives $\bar{c}^\top H = o_p(1)$ and $E^\top H = o_p(1)$, and Lemma L.2(i) gives $E^\top\bar{s} = (\bar{c}^\top E)^\top\beta^* = o_p(1)$. Hence $C_{XY} \xrightarrow{p} Q\beta^*$.

By assumption $W_N \xrightarrow{p} W_0 \succ 0$, and we have shown that $C_{XX} \xrightarrow{p} Q \succ 0$. Therefore

$$(C_{XX}^\top W_N^{-1} C_{XX})^{-1}C_{XX}^\top W_N^{-1} \xrightarrow{p} (Q^\top W_0^{-1}Q)^{-1}Q^\top W_0^{-1}.$$

Multiplying by $C_{XY} \xrightarrow{p} Q\beta^*$ yields

$$\hat{\beta}^{\text{SPLITUP}}(W_N) = (C_{XX}^\top W_N^{-1} C_{XX})^{-1}C_{XX}^\top W_N^{-1} C_{XY} \xrightarrow{p} \beta^*.$$

**L.9. Proof of Theorem 4.6**

This proof follows the same ideas as the proof of Theorem 4.2.

**Lemma L.3.** *Assume the setting of Theorem 4.6. We have*

$$\|C_{XY} - C_{XX}\beta^*\|_2 = O_p(m^{-1/2}),$$

*and consequently*

$$\left\|C_{XX}^\top W_m(C_{XY} - C_{XX}\beta^*)\right\|_\infty = O_p(m^{-1/2}).$$

*Proof.* Write $c := \mathrm{Cov}(I, X) \in \mathbb{R}^{m \times d}$ and $s := \mathrm{Cov}(I, Y) \in \mathbb{R}^m$. Set $\bar{c} := \sqrt{m}\, c$ and $\bar{s} := \sqrt{m}\, s$. Under Assumption 1, $s = c\beta^*$ and thus $\bar{s} = \bar{c}\beta^*$.

Let $\hat{c} := \sqrt{m}\, \widehat{\mathrm{Cov}}(\tilde{I}, \tilde{X})$ and $\hat{s} := \sqrt{m}\, \widehat{\mathrm{Cov}}(I, Y)$. For fold $k$, let $\hat{c}_k := \sqrt{m}\, \widehat{\mathrm{Cov}}_k(\tilde{I}, \tilde{X})$. Write the centered errors as

$$\hat{c} = \bar{c} + E, \qquad \hat{s} = \bar{s} + H, \qquad \hat{c}_k = \bar{c} + E_k.$$

Then $C_{XY} = \hat{c}^\top \hat{s}$ and $C_{XX} = \frac{1}{K(K-1)}\sum_{h \neq k} \hat{c}_h^\top \hat{c}_k$. Expanding,

$$C_{XY} = (\bar{c} + E)^\top(\bar{s} + H) = \bar{c}^\top \bar{c}\beta^* + \bar{c}^\top H + (E^\top \bar{c})\beta^* + E^\top H,$$

and

$$C_{XX} = \bar{c}^\top \bar{c} + R_m, \qquad R_m := \frac{1}{K(K-1)}\sum_{h \neq k}\left(\bar{c}^\top E_k + E_h^\top \bar{c} + E_h^\top E_k\right).$$

Therefore,

$$C_{XY} - C_{XX}\beta^* = \bar{c}^\top H + (E^\top \bar{c})\beta^* + E^\top H - R_m\beta^*.$$

By Lemma L.2(iii), $\|\bar{c}^\top H\|_2 = O_p(m^{-1/2})$ and $\|E^\top H\|_2 = O_p(m^{-1/2})$. By Lemma L.2(i), $\|\bar{c}^\top E\|_{\mathrm{op}} = O_p(m^{-1/2})$, hence

$$\|(E^\top \bar{c})\beta^*\|_2 \leqslant \|\bar{c}^\top E\|_{\mathrm{op}}\|\beta^*\|_2 = O_p(m^{-1/2})$$

since $\|\beta^*\|_2 = O(1)$. Finally, by Lemma L.2(i) and (ii), each summand in $R_m$ is $O_p(m^{-1/2})$ in operator norm, and since $K$ is fixed, the finite average satisfies $\|R_m\|_{\mathrm{op}} = O_p(m^{-1/2})$, hence $\|R_m\beta^*\|_2 = O_p(m^{-1/2})$. By the triangle inequality,

$$\|C_{XY} - C_{XX}\beta^*\|_2 = O_p(m^{-1/2}).$$

Next, since $C_{XX} \xrightarrow{p} Q$ and $W_m \xrightarrow{p} W_0$, we have $\|C_{XX}\|_{\mathrm{op}} = O_p(1)$ and $\|W_m\|_{\mathrm{op}} = O_p(1)$. Therefore,

$$\left\|C_{XX}^\top W_m(C_{XY} - C_{XX}\beta^*)\right\|_\infty \leqslant \left\|C_{XX}^\top W_m(C_{XY} - C_{XX}\beta^*)\right\|_2 \leqslant \|C_{XX}\|_{\mathrm{op}}\|W_m\|_{\mathrm{op}}\|C_{XY} - C_{XX}\beta^*\|_2 = O_p(m^{-1/2}).$$

$\square$

Let $H_m := C_{XX}^\top W_m C_{XX}$ and $b_m := C_{XX}^\top W_m C_{XY}$. The objective equals $\frac{1}{2}\beta^\top H_m \beta - b_m^\top \beta + \frac{1}{2}\|W_m^{1/2} C_{XY}\|_2^2 + \lambda_m \|\beta\|_1$. The basic inequality gives

$$\tfrac{1}{2}\Delta^\top H_m \Delta \leqslant \left\|C_{XX}^\top W_m(C_{XY} - C_{XX}\beta^*)\right\|_\infty \|\Delta\|_1 + \lambda_m(\|\beta^*\|_1 - \|\beta^* + \Delta\|_1),$$

for $\Delta := \hat{\beta}_{\ell_1}^{\mathrm{SPLITUP}} - \beta^*$. By Lemma L.3 and the choice of $\lambda_m$, with probability approaching one the score term is at most $\lambda_m/2$, yielding the cone constraint $\|\Delta_{S^{*c}}\|_1 \leqslant 3\|\Delta_{S^*}\|_1$. On this cone, $H_m \to Q^\top W_0 Q$ (by Assumption 5 (v) and fixed $d$). Assumption 5 (iv) then implies $\Delta^\top H_m \Delta \gtrsim \kappa \|\Delta\|_2^2$ for large enough $m$. Therefore $\|\Delta\|_2 \lesssim \sqrt{s^*}\lambda_m$ and $\|\Delta\|_1 \lesssim s^*\lambda_m$, proving the rates. The beta–min condition yields support recovery.

