# OpenReview forum: "Many Experiments, Few Repetitions, Unpaired Data, and Sparse Effects: Is Causal Inference Possible?"
_ICML.cc/2026/Conference — ICML 2026 spotlight_

### Official Review · Reviewer_NbxD · 2026-03-12

**Soundness:** 3
**Presentation:** 4
**Significance:** 2
**Originality:** 3
**Overall Recommendation:** 5
**Confidence:** 4

**Summary:**

The paper proposes a methodology for causal inference in a setting where treatments and outcomes are observed separately across a sample of unpaired experiments. The methodology is an extension of the traditional Instrumental variable method to the unpaired setting and it relies on a cross-fold covariance estimator plus a GMM approach. The paper shows how, without the proposed extension, traditional IV approaches are inconsistent in the unpaired setting with infinitely many experiments each with finite-samples, and, conversely the proposed method is consistent. These results are also corroborated with empirical studies.

**Compliance With Llm Reviewing Policy:**

Affirmed.

**Final Justification:**

The rebuttal offered by the authors has reinforced my initial evaluation. As such my recommendation remains that the paper be accepted.

**Key Questions For Authors:**

1. Can you provide some empirical evidence of CI coverage for the dense setting?
2. Similarly, can you provide any intuition about how one would go about quantifying uncertainty in the sparse case?
3. Can you provide some discussion or maybe even empirical evidence as to the robustness of the methodology to violations of the parametric forms for X and Y?

**Limitations:**

Yes.

**Strengths And Weaknesses:**

Soundness: The submission is generally technically sound, both from a theoretical and empirical standpoint. The main weak point is the lack of empirical testing of coverage for the asymptotic CIs derived from the results in prop. J1. As well as some discussion of ways to at least approximate variance for the estimator in the sparse setting.

Presentation: The presentation is clear and easy to follow.

Significance: The methodology is significant in the sense that the paper is convincing in arguing that the specific setting studied (many unpaired experiments) does indeed arise in certain fields. What I think hurts the broader applicability of the method is the reliance on parametric linear forms for the outcome and treatment models. Many analysts are often reluctant to make such strong assumptions of their population distributions and therefore may be discouraged from adopting the proposed method.

Originality: The work is original w.r.t. to the existing literature.

---

> ### Author Rebuttal · Authors · 2026-03-29
>
> Thank you for the thoughtful and constructive review. We appreciate your positive assessment of the technical soundness, clarity, and originality of the paper. Below we respond to your questions.
>
> **Weakness (“Significance”).** "What I think hurts the broader applicability of the method is the reliance on parametric linear forms for the outcome and treatment models. Many analysts are often reluctant to make such strong assumptions of their population distributions and therefore may be discouraged from adopting the proposed method."
>
> **Response.**
> We do not assume linearity in the treatment model. The only linearity assumption is on the structural effect of $X$ on $Y$, that is, on the causal effect itself. In particular, the dependence of $X$ on the instrument and the dependence induced by hidden confounding may still be nonlinear.
> We agree that linearity of the causal effect can be limiting in some applications. At the same time, our framework can be extended (analogously to nonlinear IV methods) to nonlinear causal effects. For example, consider a (known) feature map $\phi$ and assume $Y = \phi(X)^\top \bar\beta + \epsilon$,
> and then apply our method to the transformed covariates $\bar X \coloneqq \phi(X)$. We will add a discussion about non-linear extensions to the revised manuscript, where we explain the idea of feature transforms for extending our results.
>
> **Weakness (“Soundness”), Question 1 and Question 2.** "The main weak point is the lack of empirical testing of coverage for the asymptotic CIs derived from the results in prop. J1. As well as some discussion of ways to at least approximate variance for the estimator in the sparse setting."
> and
> "Can you provide some empirical evidence of CI coverage for the dense setting?"
> and
> "Similarly, can you provide any intuition about how one would go about quantifying uncertainty in the sparse case?"
>
> **Response.**
> We have now tested the empirical coverage of the asymptotic confidence intervals in Proposition J1 (nominal coverage is 95%). We provided them in a table below. Note that coverage increases with the sample size and that nominal coverage is reached at around $N_{total}=6400$ observations.
>
> | N_total | coverage | avg_ci_length | n_rep |
> |------------:|------------------:|-----------------------:|------:|
> |     400 |   0.3340 |          0.3936 |    500 |
> |     800 |   0.6280 |         0.3276 |      500 |
> |    1600 |   0.7740 |           0.2504 |    500 |
> |    3200 |   0.8980 |         0.1863 |       500 |
> |    6400 |   0.9440 |         0.1340 |    500 |
>
>
>
> In the sparse setting, constructing confidence intervals is difficult because the lasso is biased by the $\ell_1$ penalty. Two standard solutions are: (i) the debiased/desparsified lasso, which adds a one-step bias correction and yields asymptotically normal estimators for low-dimensional coordinates, enabling coordinate-wise CIs (Zhang and Zhang, JRSS B, 2014); and (ii) post-lasso refitting, where lasso is used for variable selection and the coefficients are then re-estimated by unpenalized least squares on the selected support, reducing shrinkage bias relative to the original lasso estimate (Belloni and Chernozhukov, 2013).
>
>
> **Question 3.** "Can you provide some discussion or maybe even empirical evidence as to the robustness of the methodology to violations of the parametric forms for X and Y?"
>
> **Response.**
> We do not assume any parametric form for $X$, and we impose only the following structural assumptions on $Y$: the causal effect of $X$ on $Y$ is linear and the error term $\epsilon$ is additive. Accordingly, there are two principal forms of misspecification to consider.
> The first is nonlinearity in the causal effect of $X$ on $Y$. Our method can be extended directly to non-linear causal effects through appropriate feature transformations, so this limitation is not fundamental.
> The second is a violation of the additive noise assumption, which can be viewed as a violation of the exclusion restriction. We will add an experiment to the revision where we show that in the finite-dimensional, dense setting small violations of the exclusion restriction (measured by $\| cov(I, \epsilon) \|$) lead to only small decreases in performance. Specifically, we plot the MAE versus total sample size for different violation strengths. The plots are provided here: https://files.catbox.moe/c6urx5.pdf,

---

> > ### Author Rebuttal · Reviewer_NbxD · 2026-04-03
> >
> > I thank the authors for engaging with my review. My score will remain unchanged.

---

> > > ### Author Response · Authors · 2026-04-08
> > >
> > > Thank you for your time and the positive feedback.

---

### Official Review · Reviewer_oyCA · 2026-03-12

**Soundness:** 4
**Presentation:** 3
**Significance:** 4
**Originality:** 4
**Overall Recommendation:** 6
**Confidence:** 4

**Summary:**

- The authors tackle the problem of cause effect estimation with unpaired data
- This is motivated by examples from biology where experiments are run multiple times with conditions and outcomes measured, unpaired
- The authors contextualise their method well in the recent two-sample IV literature, and focus on the setting with many experiments m and large dimensions d
- They provide consistency estimators that outperform TS-IV
- Synthetic and real world data experiments validate the method

**Compliance With Llm Reviewing Policy:**

Affirmed.

**Key Questions For Authors:**

- Are the results limited to linear effects as in equation 2.1?
- Are results limited to covariance assumptions 1 and 1_prime? Can those be loosened
- Fig3: n/m = 4: Why does UP-GMM track TS-IV so closely?
- Any thoughts on why UP-GMM underperforms TS-IV in Fig 5?

**Limitations:**

- Does the method assume linearity? I can’t tell if this is an actual restriction

**Strengths And Weaknesses:**

Strengths
- Great novel method building on established and well reviewed literature in MR
- Derivation is clear and simple
- Experiments are chosen well and build clear intuition and explain phenomena well were appropriate
- The use of causal chambers is an outstanding contribution and generates strong confidence through real world strong evidence


Weaknesses:
- Page 2 line 145: I am unsure if greying-out parts of equation 2.1 is the best way. Is there a color based option or a symbolic visual way to indicate unobservables?
- The peaking phenomenon observed in setting 3 raises questions, though I feel these are addressed well in the appendix, can you envision any other scenarios why we observe this behaviour?
- The authors evidently hit the page limit on page 8, with no conclusion but I think this venue allows another page post review, so do please use that for further discussion.

Decision
- given the strong theorems, great improvements over baselines and the unique experimentation with causal chambers, I recommend this paper for a highlight

---

> ### Author Rebuttal · Authors · 2026-03-29
>
> Thank you for the very positive review and for recommending the paper for highlight. We are glad that the methodological contribution, the theoretical guarantees, and the empirical evidence came through clearly. Below we respond to your questions.
>
> **Weakness 1.** "Page 2 line 145: I am unsure if greying-out parts of equation 2.1 is the best way. Is there a color based option or a symbolic visual way to indicate unobservables?"
>
> **Response.**
> Thank you for this suggestion. We did consider alternative ways of indicating unobservable quantities in Eq. 2.1, including color-based and symbolic annotations. In our view, these alternatives did not improve readability compared with the current greying-out, and in some cases made the equation more visually cluttered. We therefore kept the current notation, while remaining open to better suggestions.
>
>
> **Weakness 2.** "The peaking phenomenon observed in setting 3 raises questions, though I feel these are addressed well in the appendix, can you envision any other scenarios why we observe this behaviour?"
>
> **Response.**
> Our current interpretation is the one discussed in the appendix, and at present we do not have strong evidence for an additional mechanism beyond that explanation.
>
> **Weakness 3.** "The authors evidently hit the page limit on page 8, with no conclusion but I think this venue allows another page post review, so do please use that for further discussion."
>
> **Response.**
> We agree. In the revised manuscript we will add a dedicated discussion section that summarizes the main findings, clarifies the scope of the assumptions, and explicitly discusses limitations.
>
> **Question 1.** "Are the results limited to linear effects as in equation 2.1?" and "Does the method assume linearity? I can’t tell if this is an actual restriction."
>
> **Response.**
> As stated, our theory assumes a linear causal effect of $X$ on $Y$ as in Eq. (2.1). This linearity assumption applies only to the structural effect of $X$ on $Y$; other parts of the data-generating process may still be nonlinear. We will clarify this point in the revised manuscript and note that extending the framework to nonlinear structural effects is an interesting direction for future work.
>
> **Question 2.** "Are results limited to covariance assumptions 1 and 1 prime? Can those be loosened"
>
> **Response.** We believe that Assumption 1 cannot be substantially weakened within the present framework. Intuitively, both parts of the assumption are directly tied to identification. If Assumption 1(i), i.e., $\mathrm{cov}(I,X)=\mathrm{cov}(\tilde I,\tilde X)$, is dropped, identifiability can already fail in a simple linear-Gaussian setting; see our response to reviewer "ity6" under *Assumption 1* for an explicit counterexample. Similarly, if the exclusion restriction is violated, the instrument is no longer valid, and identification generally fails even in the paired setting. This is why we view Assumption 1 not as a cosmetic simplification, but as an essential bridge for identification in the unpaired setting.
>
> **Question 3.** "Fig3: n/m = 4: Why does UP-GMM track TS-IV so closely?"
>
> **Response.**
> Figure 3 shows the high-dimensional setting without sparsity. In this regime, the estimation error of both UP-GMM and TS-IV is mainly driven by the measurement-error bias. Since both methods are affected by the same bias mechanism, their losses track each other closely.
>
>
> **Question 4.** "Any thoughts on why UP-GMM underperforms TS-IV in Fig 5?"
>
> **Response.**
> UP-GMM underperforms TS-IV in Figure 5 because it depends on an estimated optimal weighting matrix. While this yields asymptotically efficient estimates when the sample size is large or the noise level is low, it is well-known that it can be unstable in smaller or noisier settings because the estimated weighting matrix is itself high-variance and must be inverted. This instability degrades finite-sample performance. We have now verified this hypothesis by replacing the optimal weighting matrix with the identity matrix, in which case UP-GMM performs very similarly to TS-IV. We will add a sentence to the revised manuscript.

---

> > ### Author Rebuttal · Reviewer_oyCA · 2026-04-02
> >
> > Thanks for the clarifications, no further concerns. I still recommend for highlight.

---

> > > ### Author Response · Authors · 2026-04-08
> > >
> > > Thank you for your time and for recommending the paper for highlight.

---

### Official Review · Reviewer_65wk · 2026-03-13

**Soundness:** 3
**Presentation:** 3
**Significance:** 4
**Originality:** 4
**Overall Recommendation:** 5
**Confidence:** 4

**Summary:**

To address unobserved confounding in unpaired two-sample data, the authors propose SPLITUP, a new GMM-type based sample-splitting estimator to identify the causal effect of X on Y. Traditional TS-IV methods suffer from severe attenuation bias during high-dimensional covariance estimation, and SPLITUP mitigates this bias through cross-moment construction, proving robust in high-dimensional environments and effectively accommodating sparse causal effects. Furthermore, the authors establish the exact identifiability conditions for $\beta$, establishing rigorous theoretical guarantees for the consistency of the estimator and the asymptotic normality. Finally, through synthetic simulations and a real-world Causal Chambers Light Tunnel experiment, they empirically demonstrate that SPLITUP successfully eliminates bias and maintains consistency where standard baselines fail.

**Compliance With Llm Reviewing Policy:**

Affirmed.

**Final Justification:**

The authors have answered my questions during the rebuttal. I believe that the paper is of good quality and has clear strengths. It should be accepted.

**Key Questions For Authors:**

1. Could the authors provide a sensitivity analysis to discuss its robustness against mild violations of Assumption 1?
2. Given that SPLITUP requires within-environment sample splitting, does the framework completely break down in extreme sparsity regimes with only a single observation per environment (r = 1)?
3. How does SPLITUP perform on a standard biological dataset?
4. How should practitioners systematically select the optimal regularization parameters ($\lambda_1$ and $\lambda_2$) in a purely unpaired setting where standard cross-validation is infeasible?

**Limitations:**

1. The paper’s structural organization can be improved. It can be helpful to add a Discussion or Conclusion section to synthesize the findings and limitations.
2. The proposed method's core assumption 1, which strictly enforces perfect instrument validity and exact cross-sample equivalence, is highly restrictive for high-dimensional, real-world data. The paper can be substantially improved with an additional analysis of the proposed estimator’s robustness to mild violations of these assumptions.

**Strengths And Weaknesses:**

Strengths:
1. The figures and empirical examples are presented intuitively and align perfectly with the theoretical claims.
2. The problem addressed by the proposed estimator is of immense interest in biology. Specifically, the method directly tackles the critical bottlenecks in two-sample MR, where researchers frequently grapple with high-dimensional, weak, and unconfounded but unpaired instrumental variables.
3. The authors establish a remarkably solid theoretical foundation for identifiability and consistency.
4. The experimental section effectively and thoroughly evaluates the proposed method.

Weakness:
1. The paper’s structural organization feels rushed, evidenced by orphaned subsections (e.g., Section 4.2.1) and an abrupt ending without a standard Discussion or Conclusion section to synthesize the findings and limitations.
2. The method’s reliance on Assumption 1—which strictly enforces perfect instrument validity and exact cross-sample equivalence—is highly restrictive for high-dimensional, real-world data. The paper lacks an analysis of the estimator’s robustness to mild violations of these assumptions, limiting its practical utility.

---

> ### Author Rebuttal · Authors · 2026-03-29
>
> Thank you for the positive review and for highlighting the paper’s significance, theoretical foundation, and empirical evaluation. Below we respond to your questions.
>
> **Weakness 1 and Limitations 1.** “The paper’s structural organization feels rushed, evidenced by orphaned subsections (e.g., Section 4.2.1) and an abrupt ending without a standard Discussion or Conclusion section to synthesize the findings and limitations." and "The paper’s structural organization can be improved. It can be helpful to add a Discussion or Conclusion section to synthesize the findings and limitations."
>
> **Response.**
> We thank the reviewers for this suggestion and agree that the paper would benefit from a clearer closing discussion. In the revision, we will add a dedicated discussion section that summarizes the main findings, clarifies the scope of the assumptions, and explicitly discusses limitations. Regarding Section 4.2.1, we will turn the paragraph headings into subsection titles.
>
> **Weakness 2, Limitations 2 and Question 1.** "The method’s reliance on Assumption 1—which strictly enforces perfect instrument validity and exact cross-sample equivalence—is highly restrictive for high-dimensional, real-world data. The paper lacks an analysis of the estimator’s robustness to mild violations of these assumptions, limiting its practical utility.."
> and "The proposed method's core assumption 1, which strictly enforces perfect instrument validity and exact cross-sample equivalence, is highly restrictive for high-dimensional, real-world data. The paper can be substantially improved with an additional analysis of the proposed estimator’s robustness to mild violations of these assumptions."
> and "Could the authors provide a sensitivity analysis to discuss its robustness against mild violations of Assumption 1?"
>
> **Response.** Thank you for this helpful comment. We agree that robustness to mild violations of Assumption 1 is important in practice. To address this, we will perform an additional empirical sensitivity analysis for UP-GMM and SplitUP in the dense, finite-dimensional setting. We consider two types of misspecification: (i) exclusion-restriction violations, parameterized by $cov(I,\epsilon)$, and (ii) cross-sample covariance mismatch, parameterized by $\|cov(I,X)-cov(\tilde I,\tilde X)\|$. We then plot the MAE versus total sample size for different violation strengths. We see that in the tested setting both methods are robust to small violations of Ass 1 (note that $\|cov(I,X)\| \approx 0.2$ in the setting without violations). The plots are provided here for exclusion-restriction violations: https://files.catbox.moe/c6urx5.pdf, and here for covariance mismatch: https://files.catbox.moe/mgjki7.pdf. We will revise the paper to discuss this sensitivity more explicitly.
>
> **Question 2.** "Given that SPLITUP requires within-environment sample splitting, does the framework completely break down in extreme sparsity regimes with only a single observation per environment (r = 1)?"
>
> **Response.**
> Not completely. SplitUP only splits the $(\tilde I,\tilde X)$ sample, not the $(I,Y)$ sample. So $r=1$ in $(I,Y)$ is still possible (in that the estimator is well-defined). However, if we are in the categorical-environment setting with only one $(\tilde I,\tilde X)$ observation per environment, the cross-fit denominator cannot be constructed, so the bias correction breaks down.
>
> **Question 3.** "How does SPLITUP perform on a standard biological dataset?"
>
> **Response.**
> Given that we aim to estimate a causal effect and that simultaneously intervening on many genes is not as common, the evaluation becomes non-trivial; together with our collaborators we are currently working on real-world applications and the evaluation but this will not be ready for the final version of this paper.
>
> **Question 4.** "How should practitioners systematically select the optimal regularization parameters in a purely unpaired setting where standard cross-validation is infeasible?"
>
> **Response.**
> In the revision, we will add a section discussing this issue in more detail and propose a solution. (A key point is that our use of sparsity is conceptually different from standard high-dimensional Lasso: the main issue is not a finite-sample regime with small $n$ relative to $d$, but rather a population-level identifiability issue, where sparsity is needed because the moment operator may be rank-deficient. In that sense, the problem is closer to noisy compressed sensing than to classical prediction-based model selection.) Our solution is motivated by the following fact: in the noiseless population problem, we choose the largest $\lambda$ such that the true parameter still solves the optimization problem with zero residual. We propose a noisy analogue of this idea: choose the largest $\lambda$ such that the null hypothesis that the objective is equal to zero cannot be rejected at level $\alpha$. In the revision, we will describe how to implement this via a bootstrap-based test.

---

> > ### Author Rebuttal · Reviewer_65wk · 2026-04-01
> >
> > Thank you for your response and clarification. I acknowledge your response, which fully addressed my concerns. And I understand that real-world data analysis takes time, and it may not be ready for the final version of this paper (Question 3).

---

> > > ### Author Response · Authors · 2026-04-08
> > >
> > > Thank you for your time and for the helpful feedback.

---

### Official Review · Reviewer_ity6 · 2026-03-17

**Soundness:** 3
**Presentation:** 2
**Significance:** 3
**Originality:** 3
**Overall Recommendation:** 5
**Confidence:** 2

**Summary:**

The paper studies causal effect estimation in the unpaired data setting. For example, consider biological experiments where each experimental setting has a certain set of covariates, but the setting does not independently affect the outcome beyond how it affects the covariates. In this case, as the number of experiments grows but the number of repetitions within each experiment does not, standard estimators fail to work. This paper develops a new method to tackle this and show consistency results when the number of experiments grows. The main idea (under the assumptions) is that there isn't a large enough shift between the environments to break identification with the traditional two-sample IV type method (prop 3.1 is basically invertibility), but the rest is interesting when going beyond the low-dimensional setting with sparse effect vectors and high-dimensional instruments.

**Compliance With Llm Reviewing Policy:**

Affirmed.

**Final Justification:**

The authors addressed my concerns about the novelty of the proofs

**Key Questions For Authors:**

See weaknesses.

**Limitations:**

See weaknesses.

**Strengths And Weaknesses:**

- Clean motivation of the problem
- Effective presentation, although could have been improved with a better ordering of the intro, putting the bio examples first.
- Clean algorithm via sample splitting and moment matching between the environments where Y is known and where X is known. Seems to work well across a variety of settings, especially when environments grow fast.


Weaknesses
- It seems like many of the results follow from existing results and assumption 1s, at least in the identification part. The estimation results seems novel for the GMM + sparsity part and GMM + high-dimensional part, but what's the actually novel part of the proofs? Isn't assumption 1 really just assuming away the fundamental issue that the covariances might be different, and the rest follow from the theory that holds for GMM in the paired case? Do the other moments matter for the consistency? They should because the method fully relies on covariances and variances only, and the higher-order moments may only affect the constants in the convergence rate.

- I see theorem 4.5 is novel because the rates at which environments provide more information. My main concern is how this should only work if the environments are fully independent (in some sense you get the IID assumption but with environments). Is this a reasonable view and a reasonable assumption? I see the issue as the environments growing fast means that IV gets high-dimensional and the limited repetitions per environment may not help estimate the covariances well. So it's not IID. Is this correct?

---

> ### Author Rebuttal · Authors · 2026-03-29
>
> Thank you for the thoughtful review. We appreciate that you found the problem well motivated, the results novel, and the algorithm clean. Below we respond to your questions.
>
> **Weakness 1.** "It seems like many of the results follow from existing results and assumption 1s, at least in the identification part. The estimation results seems novel for the GMM + sparsity part and GMM + high-dimensional part, what's the actually novel part of the proofs. Isn't assumption 1 really just assuming away the fundamental issue that the covariances might be different, and the rest follow from the theory that holds for GMM in the paired case?"
>
> **Response.**
> We structure our response into three categories.
>
> *Estimation:*
> Existing GMM proof techniques are insufficient to address the high-dimensional setting that we consider in this work. The key difficulty is that the first-stage cross-covariance $\widehat{cov}(\tilde I,\tilde X)$ is itself high-dimensional and can therefore not be estimated consistently when $n/m \to c \in (0,\infty)$ which prevents us from applying existing GMM results. Our main proof contribution is to realize that, even in the high-dimensional setting, the cross-product $\widehat{cov}(\tilde I,\tilde X)^{\top} \widehat{cov}(\tilde I,\tilde X)$ can be consistently estimated (under appropriate assumptions). To prove this we had to consider different moment condition and carefully consider under which conditions such convergence can be achieved.
>
> *Identifiability:* Theorem 3.2 shows that, contrary to the usual view in the two-sample IV literature, an underdetermined covariance matrix need not preclude identification under sparsity; it gives necessary and sufficient conditions for when identification holds. We therefore view it as a conceptual contribution, even though the proof uses established tools from the compressed sensing literature.
>
> *Assumption 1:*
> We do not view Assumption 1 as simply “assuming away” the core difficulty, but rather as an essential bridge for identification in the unpaired problem. If the cross-sample equality $cov(I,X)=cov(\tilde I,\tilde X)$ is dropped, identifiability can already fail in a simple linear-Gaussian setting. Indeed, one can construct two no-confounding models with the same observed distributions for $(I,Y)$ and $(\tilde I,\tilde X)$ but different causal effects $\beta^\ast$. Let $(\tilde I, \tilde X)$ be generated by the following system: $\tilde I = N(0,1)$ and $\tilde X = \tilde I + N(0,1)$. We now consider two models for $(I,X,Y)$. In the first model, $I = N(0,1)$, $X = 2 I + N(0,1)$, and $Y = X + N(0,1)$; in the second model, $I = N(0,1)$, $X = I + 1/2 N(0,1)$, and $Y = 2 X + N(0,1)$ (all noise terms are jointly independent). Both models induce the same distribution over $(I,Y)$ but the causal effects from $X$ to $Y$ (called $\beta^\ast$ in the paper) differ: in the first model $\beta^\ast =1$, in the second model $\beta^\ast =2$. Thus, we generally lose identifiability when dropping $cov(I,X) = cov(\tilde I, \tilde X)$. We will add a short remark on this after the discussion of Assumption 1.
>
>
> **Weakness 1 (cont).** "Do the other moments matter for the consistency? They should because the method fully relies on covariances and variances only, and the higher-order moments may only affect the constants in the convergence rate."
>
> **Response.**
> Higher-order moments do not play a role for identifiability.
> However, for the estimation results, our proofs use some control beyond second moments. The estimator is built from empirical covariance terms. To show that these sample moments concentrate around their population counterparts, and to establish score bounds and consistency rates we impose bounded fourth-moment assumptions (and similar regularity conditions).
>
>
> **Weakness 2.** "I see theorem 4.5 is novel because the rates at which environments provide more information. My main concern is how this should only work if the environments are fully independent [...]. Is this correct?"
>
> **Response.**
> We believe the reviewer’s intuition is correct in spirit, but we would formulate things differently. Our results do not require the environments themselves to be independent units. The independence assumption is at the level of the observations, not the environments.
> In the categorical-environment interpretation, each observation is an i.i.d.\ draw from a distribution over environments together with the corresponding variables. As $m$ grows, this becomes a growing-dimension (triangular-array) regime: the distribution changes with $m$, but within each fixed $m$ the observations are still i.i.d. So the issue is not that the data are ``not i.i.d.'' Rather, the issue is that standard fixed-dimensional IV/GMM asymptotics break down when $m$ grows with $n$ while the number of repetitions per environment stays bounded. In that regime, the covariance estimator $\widehat{cov}(\tilde I,\tilde X)$ has non-vanishing estimation error, and this creates the asymptotic bias of naive TS-IV.

---

> > ### Author Rebuttal · Reviewer_ity6 · 2026-04-04
> >
> > Thanks for the response. I'm updating my score accordingly.

---

> > > ### Author Response · Authors · 2026-04-08
> > >
> > > Thank you for your time and for the positive acknowledgement. We appreciate your feedback.

---

### Decision · Program_Chairs · 2026-04-30

**Decision:**

Accept (spotlight)

**Comment:**

In this paper, the authors study effect estimation in an unpaired two sample regime. They introduce SplitUP, a new sample splitting based Generalized Method of Moments (GMM) estimator for identifying the causal effect of X on Y. The authors focus on the setting where the number of experimental environments (perturbations) grows. While traditional methods such as TS-IV suffer from severe bias during covariance estimation and are inconsistent, SplitUP fixes it with sample splitting with consistency guarantees in both sparse and dense regimes. The observation that even though first stage cross-covariance is not consistently estimable, the cross product in the moment condition is estimable is interesting and leads to the final results. The reviewers all agree that the paper should be accepted. The responses by the authors confirmed or reinforced the positive judgement (reviewer oyCA recommended highlight) of the reviewers in all cases.